# Variational Pólya Tree

**Lu Xu**[*]
The University of Hong Kong
`xulu0908@hku.hk`

**Tsai Hor Chan**[*]
University of Pennsylvania
`Tsaihor.Chan@PennMedicine.upenn.edu`

**Kwok Fai Lam**
Hong Kong Metropolitan University
`kflam@hkmu.edu.hk`

**Lequan Yu**
The University of Hong Kong
`lqyu@hku.hk`

**Guosheng Yin**[†]
The University of Hong Kong
`gyin@hku.hk`

## Abstract

Density estimation is essential for generative modeling, particularly with the rise of modern neural networks. While existing methods capture complex data distributions, they often lack interpretability and uncertainty quantification. Bayesian nonparametric methods, especially the Pólya tree, offer a robust framework that addresses these issues by accurately capturing function behavior over small intervals. Traditional techniques like Markov chain Monte Carlo (MCMC) face high computational complexity and scalability limitations, hindering the use of Bayesian nonparametric methods in deep learning. To tackle this, we introduce the variational Pólya tree (VPT) model, which employs stochastic variational inference to compute posterior distributions. This model provides a flexible, nonparametric Bayesian prior that captures latent densities and works well with stochastic gradient optimization. We also leverage the joint distribution likelihood for a more precise variational posterior approximation than traditional mean-field methods. We evaluate the model performance on both real data and images, and demonstrate its competitiveness with other state-of-the-art deep density estimation methods. We also explore its ability in enhancing interpretability and uncertainty quantification. Code is available at `https://github.com/howardchanth/var-polya-tree`.

## 1 Introduction

Density estimation has gained significant importance in generative modeling, particularly with the advent of modern neural networks [3, 28], such as normalizing flows and autoregressive networks. These techniques have found numerous applications, including image generation and large language modeling, highlighting their versatility and effectiveness in capturing complex data distributions. While many recent efforts focus on enhancing feature learning and transforming densities [46, 9], these deep density estimation networks often struggle with interpretability, raising concerns about their trustworthiness [30].

---

[*]Equal contributions
[†]Corresponding author

39th Conference on Neural Information Processing Systems (NeurIPS 2025).

Bayesian nonparametric methods provide an appealing alternative, enabling flexible modeling that naturally adapts complexity according to data and rigorously quantifies uncertainty [32]. Incorporating Bayesian nonparametric priors such as Dirichlet processes (DP) into deep neural architectures has attracted substantial attention, yielding notable progress in regression [42] and clustering [26]. However, DP-based approaches inherently induce discreteness, making them suboptimal for directly modeling continuous distributions.

In contrast, the Pólya tree (PT) is specifically suited to continuous distributions, producing probability measures absolutely continuous with respect to the Lebesgue measure [35, 4, 29]. Yet, despite their appealing theoretical properties, PT priors have rarely been combined with modern deep learning frameworks due to computational challenges. Traditional posterior inference via Markov chain Monte Carlo (MCMC) suffers scalability limitations, becoming prohibitively expensive as data scales [48, 4].

To bridge this gap, we propose the variational Pólya tree (VPT), the first variational Bayesian integration of PT priors with deep neural networks. Our VPT leverages stochastic variational inference, enabling scalable training via stochastic gradient optimization. Importantly, our method does not rely on the simplifying independence assumptions typical of mean-field approximations. Instead, we exploit the PT prior's intrinsic hierarchical structure and conjugacy properties, allowing a tractable and exact joint posterior over Beta-distributed node weights. This approach maintains the rich dependency structure across tree nodes, facilitating efficient and precise backpropagation.

We validate VPT across diverse tasks, including high-dimensional tabular data and image density estimation using normalizing flow architectures. Unlike existing deep density estimation works that mainly focus on transformation and feature learning architectures [18, 6, 39, 15], we present a novel prior and demonstrate how VPT can enhance interpretability and provide a measure of uncertainty.

Our contributions can be summarized as follows: (1) We introduce the first variational inference framework that makes PT priors practically useful and can be used as plug-and-play components for continuous density estimation and deep generative modeling, integrating seamlessly with modern neural architectures, e.g., flows, variational autoencoders (VAEs). (2) We leverage the joint posterior likelihood as the variational objective to provide an exact approximation of the posterior distribution of the PT while minimizing computational complexity. (3) Empirical evaluations on various datasets demonstrate that our VPT prior can achieve superior performance in density estimation compared to existing methods. Additionally, ablation analysis reveals that a deeper tree generally leads to more accurate density estimation. The framework incurs minimal overhead—less than $0.05\%$ additional memory and at most $1.3\times$ runtime—while remaining end-to-end trainable with standard deep learning architectures. (4) We demonstrate that with the VPT prior, the model can effectively learn both the estimated density distributions and the associated uncertainties. Our findings also indicate that the latent features acquired using the VPT prior are significantly more meaningful than those derived from traditional distributions.

## 2 Related Works

### 2.1 Deep Density Estimation

Deep generative models have achieved significant success in density estimation, especially for high-dimensional data like images [3, 28]. These models typically transform latent variables, sampled from known distributions, into data points, enabling flexible and expressive density modeling.

Normalizing flows use sequences of invertible transformations, parameterized by neural networks, to model complex distributions. They enable efficient likelihood evaluation by maintaining tractable Jacobian computations. For example, NICE [8] uses coupling layers, while RealNVP [9] extends this with additive and multiplicative coupling layers for improved expressiveness. Autoregressive methods decompose a $D$-dimensional density into one-dimensional conditional densities, functioning as a specialized form of normalizing flow. MAF [39] uses masked neural networks to model these conditional distributions, while IAF [23] enables more efficient sampling through inverse transformations. VAEs [20] introduce stochastic transformations to learn compact representations but often suffer from posterior collapse due to the evidence lower bound (ELBO). Diffusion models [17], a hierarchical extension of VAEs, use forward noise addition and reverse denoising processes to generate data.

Despite these advances, most deep density estimators still rely on parametric assumptions and lack a principled way to encode interpretability and uncertainty about the distribution. Most recent works in this area focus on learning powerful and tractable transformations from input to the latent density. For example, Glow [22] introduces invertible $1 \times 1$ convolutions to improve expressiveness. Neural autoregressive flow (NAF) [18] employs autoregressive models within a normalizing flow framework, allowing for efficient and expressive density estimation. MuLAN [41] utilizes adaptive noise techniques within diffusion models to improve sample quality and diversity. We propose a Bayesian nonparametric prior that can be compatible with these transformation approaches, which is not only flexible to train but also provides interpretability and uncertainty quantification naturally.

## 2.2 Bayesian Nonparametric Methods

Bayesian nonparametric (BNP) models provide a framework for adapting model complexity to data [47, 40, 27, 16, 19, 43]. Classic examples include Dirichlet processes (DPs), which enable unbounded mixture components, and have been extended to deep learning, such as stick-breaking VAEs and Beta-Bernoulli processes for infinite latent features [33, 10]. However, DPs concentrate probability on discrete distributions, making them unsuitable for continuous densities without additional smoothing [35]. Variational inference for DPs also requires truncating the infinite process, which can result in biased estimates.

Pólya trees, in contrast, naturally model continuous densities by recursively partitioning the domain with random probabilities. With appropriate hyper-parameters, they place probability 1 on the space of continuous distributions [25]. Unlike DPs, PTs avoid discreteness and external smoothing. Despite this advantage, their use in modern machine learning has been limited, as existing applications relied on computationally expensive methods like MCMC [2]. Crucially, no previous work has incorporated PT priors into deep neural architectures with variational Bayes.

We address this gap by introducing a PT-based variational inference framework for deep generative models. Our proposed framework is not a plug-and-play prior placement but a principled, nontrivial integration that bridges classical Bayesian nonparametrics with modern deep learning. It leverages the distinctive theoretical properties of PTs, enabling benefits such as robustness, interpretability, generalization, and calibrated uncertainty estimation. PTs' conjugacy allows efficient updates of branch probabilities via Beta–Binomial computation, enabling a structured variational posterior that retains dependencies across the tree. Unlike mean-field methods, which assume independence [45, 11, 14], our approach preserves hierarchical coherence, avoiding oversimplified approximations.

By integrating the flexibility of BNP priors with the scalability of deep learning, our method enables efficient end-to-end training of continuous density estimators. This approach overcomes the limitations of DP-based priors for continuous data and advances adaptive, interpretable density modeling in deep generative frameworks.

## 2.3 Tree-based Models

The use of binary trees is prevalent in machine learning [7, 42, 1]. In addition to well-known methods like decision trees, random forests, boosting, and XGBoost [5], Bayesian approaches to tree models have gained popularity, particularly with advancements in computational capacity.

Further, Bayesian inference has gained enormous popularity for modeling the distribution of trees, as seen in Bayesian additive regression trees (BART) [1]. Salazar [42] introduces a variational tree model for regression that establishes a Bayesian nonparametric prior on the tree space. However, like traditional methods, it partitions the sample space rather than the probability space.

In contrast, our approach uses a PT prior to model continuous densities. A tree space that partitions the probability space would offer greater modeling flexibility, accommodating a wider range of probability measures and enabling various tasks beyond regression. Our variational algorithm preserves dependencies across tree levels, ensuring coherent uncertainty propagation and avoiding the limitations of the mean-field approximation.

# 3 Methodology

## 3.1 The Pólya Tree Prior

The PT prior is a Bayesian nonparametric approach for constructing a random probability measure. The construction is based on a recursive partition of the domain, which can be viewed as a random histogram with the sizes of the bins made sequentially smaller [38].

Consider the bins of a histogram defined by a partition of the sample space into nonempty subsets $\{B_{\epsilon_{1:L}}\}$, with subsets indexed by an $L$-digit binary number $\epsilon_{1:L} = \epsilon_1 \cdots \epsilon_L$, where each element $\epsilon_j$ ($j = 1, \ldots, L$) takes values of 0 or 1. At level $j$, we define random probabilities $P(B_{\epsilon_{1:j}})$ for each bin. Consider refining the partition by splitting each bin into *left* and *right* parts, $B_{\epsilon_{1:j}} = B_{\epsilon_{1:j}0} \cup B_{\epsilon_{1:j}1}$ and define random probabilities for the refined histogram by conditional probabilities $Y_{\epsilon_{1:j}0} = P(B_{\epsilon_{1:j}0}|B_{\epsilon_{1:j}})$ and

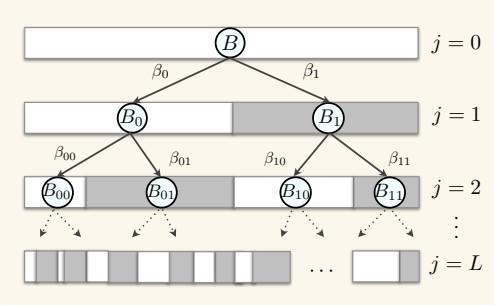

Figure 1: Graphical illustration of the Pólya tree construction. The PT prior randomly splits $B_{\epsilon_{1:j}}$ into two subintervals $B_{\epsilon_{1:j}0}$ and $B_{\epsilon_{1:j}1}$ with Beta-distributed probabilities.

$Y_{\epsilon_{1:j}1} = P(B_{\epsilon_{1:j}1}|B_{\epsilon_{1:j}})$. For any integer $L \geq 0$, this recursive refinement of bins defines a sequence of nested partitions.

A random probability measure $\mathbb{P}$ follows a PT distribution $\text{PT}(\mathcal{A}, \{\beta_j\}_{j=1}^{L})$ with parameters $\mathcal{A} = \{\alpha_\epsilon\}$ on the sequence of partitions $\{\beta_j\}_{j=1}^{L}$ where $\alpha_\epsilon$ refers to the collection of Beta distribution random variables (i.e., the branching probabilities), if there exist splitting probabilities $0 \leq Y_\epsilon \leq 1$ such that for any $\epsilon$: (1) The variables $Y_{\epsilon 0}$ are mutually independent and the posterior follow a $\text{Beta}(\alpha_{\epsilon_0}, \alpha_{\epsilon_1})$ distribution; (2) $Y_{\epsilon_1} = 1 - Y_{\epsilon_0}$; (3) For any $L \geq 0$, $P(B_{\epsilon_{1:L}}) = \prod_{j=1}^{L} Y_{\epsilon_1 \ldots \epsilon_j}$.

Figure 1 illustrates this recursive representation. In summary, the PT prior defines a random probability distribution on distributions on $[0, 1]$ by assigning any partitioning subset $B_\epsilon$ with probability $P(B_\epsilon) = \prod_{j=1, \epsilon_j=0}^{L} Y_{\epsilon_1 \ldots \epsilon_{j-1}0} \times \prod_{j=1, \epsilon_j=1}^{L} (1 - Y_{\epsilon_1 \ldots \epsilon_{j-1}0})$. Algorithm 1 shows the overall workflow of our VPT.

## 3.2 Variational Pólya Tree

**Variational Objective**. Our VPT approach leverages the intrinsic hierarchical structure of the Pólya tree to avoid traditional independence assumptions common in mean-field variational inference. Specifically, we parameterize the tree partitions $\{\beta_j\}_{j=1}^{L}$ by learnable neural networks. Due to the conjugacy of the PT prior, the posterior retains the PT form, allowing closed-form updates for each node's Beta parameters. This structural property preserves hierarchical dependencies among latent variables without any simplifying assumptions, which distinguishes VPT from the classical mean-field approximation.

Let $\mathbf{x} = \{x_i\}_{i=1}^{N}$ be the observed data, $\{\beta_{i,j}\}_{j=1}^{L}$ be the $L$-level tree partition proba-

---

**Algorithm 1** Training Procedure of Variational Pólya Tree (VPT)

---

1: **Input:** Data points $\{x_i\}_{i=1}^{N}$, dimension $D$, tree level $L$, number of epochs $n_{\text{epoch}}$, learning rate $\eta$.
2: Compute total number of nodes per dimension: $n_{\text{nodes}} = 2^L - 1$
3: Initialize Beta parameters for all nodes: $\alpha_{\epsilon_{1:j-1}0} = 1, \alpha_{\epsilon_{1:j-1}1} = 1$
4: **for** epoch $= 1, \ldots, n_{\text{epoch}}$ **do**
5:     **for** each dimension $d = 1, \ldots, D$ **do**
6:         **for** each node $j = 1, \ldots, n_{\text{nodes}}$ **do**
7:             Sample split:
8:             $Y_{\epsilon_{1:j-1}0}^{(d)}|x \sim \text{Beta}\big(\alpha_{\epsilon_{1:j-1}0}^{(d)}, \alpha_{\epsilon_{1:j-1}1}^{(d)}\big)$
9:             Compute partition intervals $B_{\epsilon_{1:j}}^{(d)}$.
10:         **end for**
11:     **end for**
12:     Compute joint posterior in Eq. (1),
13:     $p(\{\beta_j\}_{j=1}^{L}, \mathcal{Y}^L \mid \mathbf{x})$, and
14:     $\mathcal{L}_{\text{VPT}} = \log p(\{\beta_j\}_{j=1}^{L}, \mathcal{Y}^L \mid \mathbf{x})$
15:     Update $\alpha_{\epsilon_{1:j-1}0}, \alpha_{\epsilon_{1:j-1}1}$.
16: **end for**

---

bilities corresponding to the $i$-th observation, and $\mathcal{Y}^L$ be the set of all $Y_{\epsilon_{1:j}}$ in the tree. Following

Paddock et al. [38], the joint posterior distribution of an $L$-level tree given $\mathbf{x}$ can be written as

$$p(\{\beta_j\}_{j=1}^L, \mathcal{Y}^L|\mathbf{x}) = \prod_{i=1}^N \frac{1}{\nu(B_{\epsilon_{i,1:L}})} \prod_{j=1}^L Y_{\epsilon_{1:j}} p(\beta_{i,j}) \prod_{\forall\{\epsilon_{1:j-1}\}} Y_{\epsilon_{1:j-1}0}^{(\alpha_{\epsilon_{1:j-1}0}-1)} Y_{\epsilon_{1:j-1}1}^{(\alpha_{\epsilon_{1:j-1}1}-1)}, \quad (1)$$

where $\nu(B_{\{\epsilon_{i,1:L}\}}) = \prod_{j=1}^L \beta_{i,j}^{1-\epsilon_{i,j}}(1-\beta_{i,j})^{\epsilon_{i,j}}$ and $p(\beta_{i,j})$ is the prior distribution for $\beta_{i,j}$. Data points $x_i$ influence densities through two components, the probabilities $B_{\epsilon_{1:j}}$ indicating the likelihood of $x_i$ falling into a specific subset, and the conditional probabilities $Y_{\epsilon_{1:j}}$ that govern the likelihood of $x_i$ in child subsets given its parent subset. The gradients can be computed and backpropagated to update the parameters $\alpha_{\epsilon_{1:j-1}0}$ and $\alpha_{\epsilon_{1:j-1}1}$ of the Beta distributions within the Pólya tree.

We optimize VPT parameters by maximizing the log-likelihood (i.e., the ELBO) $\mathcal{L}_{\text{VPT}} = \log p(\mathbf{x}|\{\beta_j\}_{j=1}^L, \mathcal{Y}^L)$. Note that our method does not explicitly include a Kullback–Leibler (KL) divergence term, and the corresponding entropy of the posterior acts as an implicit regularization. This entropy term, which appears in the ELBO, plays a similar role by controlling the spread of the variational distribution and balancing data fit versus uncertainty. Nevertheless, incorporating an explicit KL (as shown in Appendix A) or entropy term can further balance exploration of the prior.

**Density Estimation with Flow-based Models.** The VPT prior alone defines a flexible density over a sample space. For complex high-dimensional data, we can further enhance its flexibility by embedding the VPT with deep neural networks. Taking flow-based networks as an example, we use the PT density as the base distribution and apply an invertible deep transformation $f$ from the latent space to the observed data space. Formally, let $\mathbf{x}$ represent samples drawn from an unknown $D$-dimensional distribution. The goal of density estimation is to model the underlying distribution function $F(\cdot)$. Instead of assuming a fixed parametric form for $F(\cdot)$ as in the previous work [42], we follow a Bayesian nonparametric approach by placing a VPT prior on $F(\cdot)$, thereby treating the distribution itself as random. Consider a bijective function $f : \mathcal{X} \to \mathcal{Z}$ with the inverse function $f^{-1}$ and Jacobian $J_f(x)$. Under this transformation, the density in the latent space $\mathcal{Z}$ can be computed as

$$p_Z(z) = p_X(f^{-1}(z)) \left| \det\left(J_f(f^{-1}(z))\right)\right|,$$

where $p_X(x)$ is modeled by VPT prior. The determinant of the Jacobian ensures the validity of the density transformation, which can be computed following Dinh et al. [8]. The model is optimized by maximizing the log-likelihood: $\mathcal{L} = \log p_Z(z)$. Unlike the standard flows with fixed base distributions, VPT base distribution is learned and updated via the posterior of the PT. In practice, we alternate analytic posterior updates of the PT parameters with gradient-based optimization.

### 3.3 Computation of Intervals

We use a tree data structure to compute the intervals of each leaf node [38]. For the $i$-th sample $x_i$, we first sample $\{\beta_{i,j}\}_{j=1}^L$ from its variational posterior distribution. For dimension $d$ where $x_i^{(d)} \in \mathbb{R}$, the interval $(0, 1]$ is split into two subintervals with probabilities $\beta_0^{(d)}$ and $1 - \beta_0^{(d)}$, so that the subinterval $B_0^{(d)}$ is of length $\beta_1^{(d)}$ and $B_1^{(d)}$ is of length $1 - \beta_1^{(d)}$. Next, each of $B_0^{(d)}$ and $B_1^{(d)}$ is further split into two subintervals where the splitting probability depends on $\beta_2^{(d)}$ as follows,

$$B_{00}^{(d)} = (0, \beta_1^{(d)}\beta_2^{(d)}), \quad B_{01}^{(d)} = (\beta_1^{(d)}\beta_2^{(d)}, \beta_1^{(d)})$$

$$B_{10}^{(d)} = (\beta_1^{(d)}, \beta_1^{(d)} + (1-\beta_1^{(d)})\beta_2^{(d)}), \quad B_{11}^{(d)} = (\beta_1^{(d)} + (1-\beta_1^{(d)})\beta_2^{(d)}, 1].$$

The intervals at level $j$ are obtained iteratively by applying the above rule.

### 3.4 Posterior Inference and Interpretability

With the learned $\{\beta_{i,j}\}_{j=1}^L$ and $\{\alpha_j\}_{j=1}^L$ for the $i$-th sample $x_i$ with $D$ dimensions, we can sample from the Beta distribution to obtain the probability of splits. Due to the conjugacy of the PT prior, the posterior predictive distribution has an intuitive form,

$$p(x \mid \mathcal{Y}^L) = \frac{1}{\nu(B_L(x))} \prod_{d=1}^D \prod_{j=1}^L \left(Y_{s_j^{(d)}(x)\,0}^{(d)}\right)^{1-\epsilon_j^{(d)}(x)} \left(1 - Y_{s_j^{(d)}(x)\,0}^{(d)}\right)^{\epsilon_j^{(d)}(x)}.$$

where $\mathcal{Y}^L = \{Y^{(d)}_{\epsilon_{1:j-1}\,0}\}$, $Y^{(d)}_{\epsilon_{1:j-1}\,0}$ is the (left) split variable at node $\epsilon_{1:j-1}$ in dimension $d$, and under the variational posterior $Y^{(d)}_{\epsilon_{1:j-1}\,0} \mid \mathbf{x} \sim \text{Beta}\big(\alpha^{(d)}_{\epsilon_{1:j-1}0}, \alpha^{(d)}_{\epsilon_{1:j-1}1}\big)$ with learned shape parameters $\alpha^{(d)}_{\epsilon_{1:j-1}0}, \alpha^{(d)}_{\epsilon_{1:j-1}1}$, $s^{(d)}_j(x) = \epsilon^{(d)}_{1:j-1}(x)$ is the parent index used at level $j$, $B_L(x) = \prod_{d=1}^{D} B^{(d)}_{\epsilon^{(d)}_{1:L}(x)}$ is the depth-$L$ leaf containing $x$, and $\nu\big(B_L(x)\big) = \prod_{d=1}^{D}\big|B^{(d)}_{\epsilon^{(d)}_{1:L}(x)}\big|$ is its volume.

This hierarchical organization of posterior parameters makes the model highly interpretable. One can traverse the tree to see how the model allocates probability at different scales and locations. Each level of the tree gives a coarse-to-fine view of the density.

**Uncertainty quantification.** As a Bayesian nonparametric method, one of the key advantages of our VPT is to learn a distribution over the density estimation. This allows us to obtain the variance of the learned model. Given a trained $L$-level VPT, the posterior variance can be modeled as the mean over the variances of terminal node distributions, where the posterior variance of each subinterval can be computed easily from the Beta distribution.

## 4  Experiments

We evaluate VPT on multiple density estimation tasks. We first demonstrate the qualitative benefits of VPT compared to traditional Gaussian and logistic priors using simple synthetic data. We then assess VPT quantitatively on UCI and image datasets. Our findings indicate that models with VPT priors not only achieve superior likelihood results but also provide robust uncertainty estimates.

**Implementation details**. Training a VPT involves optimizing a set of Beta distribution parameters. To ensure the Beta distribution parameters remain positive, we reparameterize them via a $\text{softplus}(\cdot)$ transformation. To integrate VPT prior seamlessly with standard generative networks, we use a sigmoid layer to project latent variables into the interval $[0, 1]$. This sigmoid mapping is computationally efficient, adding minimal complexity since its Jacobian determinant can be computed trivially.

### 4.1  Density Estimation with 2D Synthetic Data

We first illustrate the capability of our VPT prior using three common synthetic datasets, a ring of 8 Gaussians, two interwoven spirals, and a checkerboard pattern. For these experiments, we adopt a simple block neural autoregressive flow (Block-NAF) architecture as the feature learning backbone, specifically employing one flow layer consisting of two hidden layers, each with 50 units. This modest architecture choice is intentional, as our primary goal is not to achieve state-of-the-art performance on synthetic benchmarks, but rather to clearly demonstrate the benefit of the VPT prior compared to standard priors (Gaussian and logistic) under identical model complexity.

In Figure 2, we compare the learned densities using three different priors, a 2-level VPT, a 3-level VPT, and an isotropic Gaussian. While the isotropic Gaussian prior allows the model to approximate the general shape of multimodal distributions, it struggles to capture sharp boundaries and disconnected regions accurately. In contrast, both the 2-level and 3-level VPT priors clearly yield more precise representations, effectively capturing regions of low density and the multimodal structure inherent in the data. Moreover, increasing the level of the VPT prior from two to three levels further enhances its modeling capacity, resulting in notably improved density estimation and clearer separation among data clusters.

Figure 2: Density estimation results of an isotropic Gaussian prior, a 2-level VPT, and a 3-level VPT with 2D synthetic datasets.

Table 1: Log-likelihoods (standard deviations) on the test sets for density estimation using our VPT and baseline methods on real datasets with 5 runs. Higher values indicate better estimation.

| Methods | POWER | GAS | HEPMASS | MINIBOONE | BSDS300 |
|---|---|---|---|---|---|
| Real NVP [9] | $0.17_{(0.01)}$ | $8.33_{(0.14)}$ | $-18.71_{(0.02)}$ | $-13.55_{(0.49)}$ | $153.28_{(1.78)}$ |
| Glow [22] | $0.17_{(0.01)}$ | $8.15_{(0.40)}$ | $-18.92_{(0.08)}$ | $-11.35_{(0.07)}$ | $155.07_{(0.03)}$ |
| MADE MoG [39] | $0.40_{(0.01)}$ | $8.47_{(0.02)}$ | $-15.15_{(0.02)}$ | $-12.27_{(0.47)}$ | $153.71_{(0.28)}$ |
| FFJORD [15] | $0.46_{(0.01)}$ | $8.59_{(0.12)}$ | $-14.92_{(0.08)}$ | $-10.43_{(0.04)}$ | $157.40_{(1.78)}$ |
| MAF MoG [39] | $0.30_{(0.01)}$ | $9.59_{(0.02)}$ | $-17.39_{(0.02)}$ | $-11.68_{(0.44)}$ | $156.36_{(0.28)}$ |
| TAN [36] | $0.60_{(0.01)}$ | $12.06_{(0.02)}$ | $\mathbf{-13.78}_{(0.02)}$ | $-11.01_{(0.48)}$ | $159.80_{(0.07)}$ |
| NAF-DDSF [18] | $0.62_{(0.01)}$ | $11.96_{(0.33)}$ | $-15.09_{(0.40)}$ | $-8.86_{(0.15)}$ | $157.43_{(0.30)}$ |
| Block-NAF [6] | $0.57_{(0.01)}$ | $11.01_{(0.10)}$ | $-15.06_{(0.08)}$ | $-8.90_{(0.31)}$ | $156.33_{(0.81)}$ |
| VPT ($L=4$) | $\mathbf{0.67}_{(0.01)}$ | $11.92_{(0.10)}$ | $-15.01_{(0.04)}$ | $-8.71_{(0.25)}$ | $158.59_{(1.02)}$ |
| VPT ($L=6$) | $0.61_{(0.01)}$ | $\mathbf{12.20}_{(0.06)}$ | $-13.94_{(0.05)}$ | $\mathbf{-8.51}_{(0.03)}$ | $\mathbf{162.72}_{(0.94)}$ |

## 4.2 Density Estimation with Real Data

We perform density estimation on five tabular UCI datasets, POWER, GAS, HEPMASS, MINI-BOONE, and BSDS300. Detailed information about these datasets is summarized in Appendix B (Table A1). We follow the preprocessing procedure outlined in [39]. For this experiment, we employ a Block-NAF [6] as our feature learning architecture, which has fewer parameters compared to the original neural autoregressive flow [18]. Consistent with the Block-NAF methodology, we train 5 stacked flows, each with 2 layers and $20D$ hidden units, where $D$ represents the input dimension. The model is trained using the Adam optimizer, with a learning rate of $10^{-2}$ for the Block-NAF flow and 0.1 for the variational Pólya tree.

**Results**. Table 1 summarizes the reported log-likelihood values. With a 4-level VPT, our model outperforms Block-NAF across all datasets. Additionally, with a 6-level VPT, our model demonstrates an even greater margin of improvement over Block-NAF, except on the POWER dataset.

By comparing our results with deeper density estimation models, we observe that our VPT models achieve the state-of-the-art performance across various datasets, with the exception of the HEPMASS dataset, where we attain results comparable to TAN [36].

Table 2: Number of model parameters, relative to Block-NAF. Block-NAF and NAF use Gaussian priors, while VPTs with $L=4$ or 6 use the Block-NAF backbone with the VPT prior. The notation "×" indicates a multiplicative factor, while "+" indicates an additive increase.

| Datasets | Block-NAF | NAF | VPT(4) | VPT(6) |
|---|---|---|---|---|
| POWER | $414,213$ | $\times 4.57$ | $+180$ | $+756$ |
| GAS | $401,741$ | $\times 2.60$ | $+240$ | $+1008$ |
| HEPMASS | $9,272,743$ | $\times 35.88$ | $+630$ | $+2646$ |
| MINIBOONE | $7,487,321$ | $\times 87.91$ | $+1290$ | $+5418$ |
| BSD300 | $36,759,591$ | $\times 16.48$ | $+1890$ | $+7938$ |

**Number of parameters**. We compare model complexity against NAF [18] and Block-NAF [6]. Our VPT adds only $(2^L - 1) \times 2 \times D$ parameters to Block-NAF, a negligible overhead compared to NAF. Despite this minimal increase, VPT significantly surpasses Block-NAF in performance, demonstrating superior modeling power and efficiency (see Table 2).

## 4.3 Image Density Estimation and Image Generation

We further test our method on two image datasets: MNIST and CIFAR-10 [24]. We employ a classic flow-based network NICE [8] as our feature learning backbone, and we use the same settings as in the original paper. Additional details of implementations are presented in the Appendix.

**Results**. In Figure 3, we compare our VPT method with the vanilla NICE method [8] using two priors, Gaussian and logistic distributions. It is important to note that the metric employed in the original paper is log-likelihoods in the logit space, which is not directly comparable across different

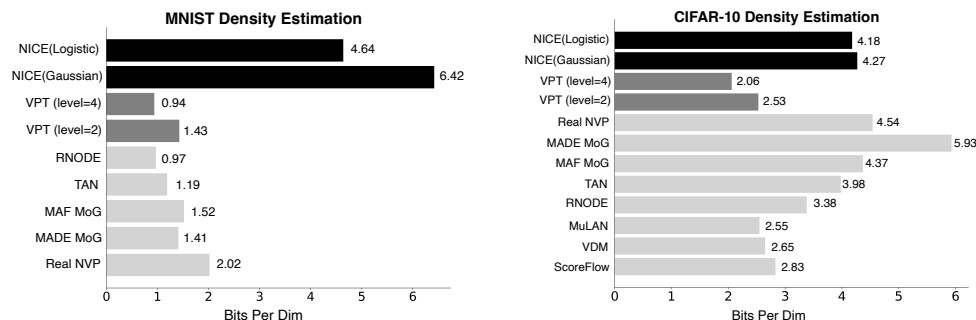

Figure 3: Negative log-likelihood in bits-per-dimension on the MNIST and CIFAR-10 test sets. Our method, VPT (shown in gray), shares the same architecture as NICE (shown in black). The results from other methods (shown in light gray) are sourced from their respective papers.

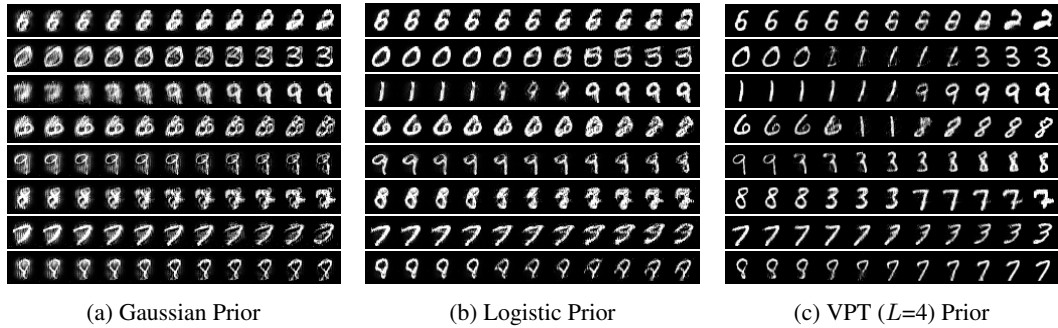

(a) Gaussian Prior        (b) Logistic Prior        (c) VPT ($L$=4) Prior

Figure 4: Interpolation on the MNIST images. While with the Gaussian prior and logistic prior the latent spaces are focused on the pixel-level features, the mixed latent features with our VPT prior reveal how the particular tree nodes are structured.

estimation methods. Therefore, we re-implemented their experiments and report the normalized negative log-likelihood in bits-per-dimension [39]: $-\frac{1}{D}\log_2 p(x)$. Our results indicate that the VPT prior significantly enhances the likelihood for both datasets.

We also compare our results with a variety of density estimation methods in Figure 3. These include flow-based methods such as Real NVP [9], MADE MoG [39], and MAF MoG [39]; RNN-based transformation approaches like TAN [36]; the Neural ODE method (RNODE) [12]; and more recently developed diffusion-based methods, including ScoreFlow [44], VDM [21], and MuLAN [41]. We observe that our VPT prior performs best among all the flow-based density estimation methods, while demonstrating comparable performance with the recently developed diffusion-based methods. With the 4-level VPT, our VPT achieves a negative log-likelihood of $0.94$ on MNIST, which is similar to $0.97$ by RNODE. On the CIFAR-10 data, our results with the 2-level VPT are comparable to those of MuLAN, while the performance can be further enhanced with the 4-level VPT.

### 4.4 Uncertainty Analysis and Interpretation

**Uncertainty analysis**. We compare uncertainty calibration with other uncertainty estimation methods, MC-dropout [13] and variational Bayesian neural network (BNN) with mean-field Gaussian weights [34], using the standardized squared error (SSE). For every sample and every dimension $d = 1, \ldots, D$, we compute SSE $= \sum_{i=1,d=1}^{N,D} \{z_i^{(d)}\}^2/ND$, where $z_i^{(d)} = (x_i^{(d)} - \mu^{(d)})/\sigma^{(d)}$. If the predictive variance is perfectly calibrated then $\mathbb{E}(z) = 1$, thus SSE $= 1$, otherwise SSE $> 1$ indicates underestimated variance, SSE $< 1$ indicates over-estimated variance.

In Table 3, the SSE results show that VPT slightly over-estimates its variance, while keeping the best calibration. MC-dropout and BNN underestimate their variances. The hierarchical structure of VPT updates the shrunk leaf probabilities toward their parents when data are sparse. This automatically inflates predictive variance in low-count regions. MC-Dropout [13] and mean-field BNNs [34]

variance often fails to fully propagate to the output, leading to the systematic under-estimation.

Figure 5 visualizes posterior variance estimates from our trained 4-level VPT on MNIST. Higher uncertainty is observed around the central digit region, aligning well with intuition—this area exhibits greater variability across digits. This confirms our model's capability to provide meaningful uncertainty quantification alongside accurate density estimation.

We emphasize that VPT's variances are analytic, no Monte-Carlo sampling is required at test time, whereas both baselines need extra computation per input. Thus VPT offers better calibration without extra computational burden.

**Effectiveness of the tree structure**. To evaluate the effectiveness of the hierarchical structure of VPT, we compare it with a learnable histogram (LH) prior. We set the number of bins to $K = 2^L$ to match the number of nodes in an $L$-level VPT. For each dimension $d$, we parametrize a increasing sequence $b_0^{(d)} < b_1^{(d)} < \cdots < b_K^{(d)}$, by setting the lower boundary $b_0^{(d)}$ and $b_k^{(d)} = b_{k-1}^{(d)} + \mathrm{softplus}(\delta_{k-1}^d)$. The parameters $\delta_{k-1}^d$ are learned jointly with the backbone network, and $\mathrm{softplus}(\cdot)$ guarantees positive bin widths. Given one dimension $x^{(d)}$, we can find the active bin index $k_*$, and compute the log-likelihood as $\log p(x^{(d)}) = \log p_{k_*}^{(d)} - \log(b_{k_*+1}^{(d)} - b_{k_*}^{(d)})$. All the hyperparameters are kept the same during training as those in VPT.

Table 3: Predictive variance calibration with SSE closer to 1 the better.

| Datasets | VPT | MC-Dropout | BNN |
|---|---|---|---|
| POWER | 0.92 | 1.30 | 1.17 |
| GAS | 0.90 | 1.35 | 1.14 |
| MNIST | 0.92 | 1.27 | 1.08 |

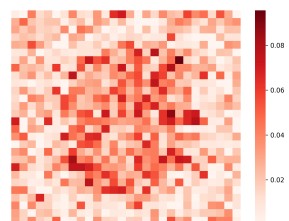

Figure 5: Posterior variance of the trained 4-level VPT model on the MNIST dataset. On each dimension/pixel ($d = 1, \ldots, D$, where $D = 28 \times 28$), we visualize the mean over the variances of terminal node Beta distribution.

Table 4 shows the results on UCI datasets with the Block-NAF backbone. The LH prior is used as a plug-in. We observe a clear advantage of VPT over LH, although they share the same number of parameters. We compare LH with VPT on CIFAR-10 with the NICE network. With $L = 4$, VPT obtains a result of 2.06 bits-per-dimension, and LH yields 2.57 BPD.

Table 4: Log-likelihoods (standard deviations) on the test set for density estimation using our VPT and the learnable histogram (LH) prior under the same setting ($L = 4$ and $K = 2^4$) on three real datasets with 5 runs. Higher values indicate better estimation.

| Priors | POWER | GAS | BSD300 |
|---|---|---|---|
| Learnable histogram (LH) | $0.62_{(0.01)}$ | $11.00_{(0.20)}$ | $156.98_{(0.93)}$ |
| Variational Pólya tree (VPT) | $0.67_{(0.01)}$ | $11.92_{(0.10)}$ | $158.59_{(1.02)}$ |

While a truncated factorized VPT may resemble a learnable histogram in terms of producing a piecewise-constant density, VPTs are more robust. Unlike an LH, where bins are estimated independently, VPT leverages the hierarchical conjugacy, enabling information sharing across scales. Specifically, every Beta-distributed node in the VPT is coupled to its ancestors. This hierarchical shrinkage allows small bins with few data points to borrow statistical strength from parent nodes, automatically adjusting the degree of smoothing. The VPT's variational posterior inference maintains the joint posterior over tree splits. This enables a more coherent and uncertainty-aware density estimate, especially in regions of sparse data. In contrast, LHs often suffer from overfitting in low-density regions or collapsing of underpopulated bins.

**Interpretation of VPTs**. Besides density estimation, VPT naturally provides probability mass at multiple scales, providing 'coarse-to-fine' insights. To better understand what the model has learned, we perform interpolation by linearly mixing the latent representations of MNIST images. Figure 4 illustrates the interpolation paths between pairs of digits under different prior distributions.

Compared to Gaussian and logistic priors, VPT yields meaningful traversals in the latent space. For example, when interpolating from '0' to '3', the intermediate representations pass through recognizable '1'. Similarly, interpolating from '9' to '8' reveals a transition through '3'. These patterns suggest that, under the VPT prior, the latent space is hierarchically organized, a node

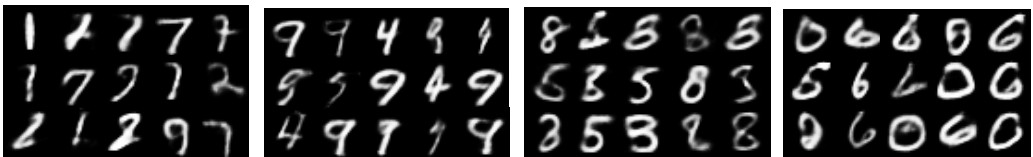

Figure 6: From left to right: images sampled from leaf nodes (nodes $3, 4, 5, 6$) of a 3-level VAE-VPT.

corresponding to the digit '1' likely resides between nodes representing '0' and '3', while a node associated with '3' is positioned between those representing '9' and '8'. This hierarchical structure offers an interpretable organization of latent representations compared to standard unimodal priors.

## 5 VAEs with the VPT Prior

Beyond the above experiments, we demonstrate the flexibility of our VPT prior by integrating it into a VAE. We use a vanilla VAE architecture with fully connected encoder and decoder networks, each consisting of two hidden layers of $400$ neurons and a latent space dimension of $2$. By simply replacing the Gaussian prior with our VPT prior, we define a modified ELBO objective as

$$\mathcal{L}_{\text{VPT-VAE}} = \mathbb{E}_{q_\phi(z|x)}[\log p_\theta(x|z)] - \lambda \sum_{v \in \text{nodes}} \text{KL}(q_\phi(z_v|x) \| p(z_v)),$$

where the first term is the reconstruction error as in the standard VAE and each node $v$ has a KL term comparing the approximated posterior $q_\phi(z_v|x)$ to the VPT prior $p(z_v)$.

Specifically, we implement a 3-level VPT prior using a tree with 7 nodes, where node 0 is the root, nodes 1–2 are intermediate nodes, and nodes 3–6 are leaf nodes. After training, we generate images by directly sampling latent variables conditioned on each leaf node (see Figure 6). The leaf nodes naturally cluster visually similar digit images. For instance, node 3 generates digit images resembling '1', '2' and '7', while node 4 produces '9' and '4'. Leaf node 5 tends to generate curved digits like '3', '5' and '8', and node 6 generates looped digits such as '0' and '6'. This demonstrates that the hierarchical partitioning induced by the VPT prior effectively captures meaningful groupings within the latent space, which enhances the interpretability compared to a VAE with a Gaussian prior.

## 6 Conclusion

We introduce the variational Pólya tree, the first deep generative framework to integrate continuous Pólya tree priors with neural architectures. VPT effectively captures complex, high-dimensional densities and scales to large datasets. While classical BNP models such as Mondrian forests [31] adaptively grow deeper with more data to guarantee universal consistency, we compensate for the fixed tree level $L$ using expressive neural transformations that allow shallow trees to capture complex structures in latent space. Although a fixed $L$ may limit asymptotic consistency, our approach maintains key theoretical properties. As supported by Orbanz [37], a truncated PT can still achieve local $\alpha$-Hölder continuity at non-dyadic points, preserving approximation capability without requiring strong continuity assumptions. Furthermore, we learn the Beta parameters via variational inference rather than fixing them the priori, enhancing adaptivity while preventing overfitting.

Extensive experiments show that VPT significantly improves density estimation performance across various datasets. Beyond enhanced predictive accuracy, our approach provides meaningful uncertainty quantification and improved interpretability. Our findings demonstrate that VPT delivers competitive performance at different levels of the tree. This research opens promising avenues for advancing density estimation models using Bayesian nonparametric methods.

## Acknowledgements

This work was supported in part by the Research Grants Council of Hong Kong (27206123, 17200125, C5055-24G, and T45-401/22-N), the Hong Kong Innovation and Technology Fund (GHP/318/22GD), the National Natural Science Foundation of China (No. 62201483), and the Guangdong Natural Science Fund (No. 2024A1515011875).

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

# A  Distributions and KL-Divergences

**Beta distribution.** Let $X \sim \text{Beta}(\alpha, \beta)$ and $Y \sim \text{Beta}(\gamma, \delta)$ with $\alpha, \beta, \gamma, \delta > 0$, and let $p$ and $q$ denote their densities. The KL-divergence between the two Beta distributions is

$$\text{KL}(p \,\|\, q) = \log \frac{B(\gamma, \delta)}{B(\alpha, \beta)} + (\alpha - \gamma)\big[\psi(\alpha) - \psi(\alpha + \beta)\big] + (\beta - \delta)\big[\psi(\beta) - \psi(\alpha + \beta)\big],$$

where $B(\cdot, \cdot)$ is the Beta function and $\psi(\cdot)$ is the digamma function.

For $X \sim \text{Beta}(\alpha, \beta)$, the variance is

$$\text{Var}[X] \;=\; \frac{\alpha\,\beta}{(\alpha + \beta)^2(\alpha + \beta + 1)}.$$

**Multivariate Gaussian Distribution**

The density of a multivariate Gaussian distribution is defined as

$$p(\mathbf{x}; \boldsymbol{\mu}, \boldsymbol{\Sigma}) = \frac{1}{(2\pi)^{\frac{p}{2}} |\boldsymbol{\Sigma}|^{\frac{1}{2}}} \exp\left\{ -\frac{1}{2}(\mathbf{x} - \boldsymbol{\mu})^\top \boldsymbol{\Sigma}^{-1}(\mathbf{x} - \boldsymbol{\mu}) \right\},$$

where $\boldsymbol{\mu} \in \mathbb{R}^p$ is a $p$-dimensional mean vector and $\boldsymbol{\Sigma} \in \mathbb{R}^{p \times p}$ is the covariance matrix. The KL-divergence between two multivariate normal distributions $\mathcal{N}(\boldsymbol{\mu}_1, \boldsymbol{\Sigma}_1)$ and $\mathcal{N}(\boldsymbol{\mu}_2, \boldsymbol{\Sigma}_2)$ is

$$\text{KL}(\mathcal{N}(\boldsymbol{\mu}_1, \boldsymbol{\Sigma}_1) \| \mathcal{N}(\boldsymbol{\mu}_2, \boldsymbol{\Sigma}_2)) = \frac{1}{2}\Big[ \log \frac{|\boldsymbol{\Sigma}_2|}{|\boldsymbol{\Sigma}_1|} - p + \text{tr}\{\boldsymbol{\Sigma}_2^{-1}\boldsymbol{\Sigma}_1\} + (\boldsymbol{\mu}_2 - \boldsymbol{\mu}_1)^\top \boldsymbol{\Sigma}_2^{-1}(\boldsymbol{\mu}_2 - \boldsymbol{\mu}_1) \Big].$$

# B  Experiment Details

The VPT is implemented with *PyTorch*. All experiments are conducted on a single RTX-3090 GPU.

**Density estimation with real data**.

The details of the dataset used in this experiment are summarized in Table A1.

In this experiment, we use Block-NAF [6] as the backbone feature learning network for VPT. We train the backbone network using Adam with Polyak averaging and apply an exponentially decaying learning rate schedule, starting at the learning rate of $10^{-2}$ with a decay rate of $\lambda = 0.5$ and a patience of 20 epochs for no improvement. In contrast, our VPT is trained with a constant learning rate of $0.1$. All models are trained until convergence, with a maximum of $1,000$ epochs, stopping if there is no improvement on the validation set for 100 epochs.

Table A1: Summary of five real UCI datasets.

| Datasets | Dimensions | No. Samples |
|---|---|---|
| POWER | 6 | $2,049,280$ |
| GAS | 8 | $1,052,065$ |
| HEPMASS | 21 | $525,123$ |
| MINIBOONE | 43 | $36,488$ |
| BSDS300 | 63 | $1,300,000$ |

**Density estimation and image generation.** We use NICE [8] as the backbone network for this experiment, employing a dequantized version of the data. The architecture consists of a stack of four coupling layers, with a diagonal positive scaling for the last stage. Each coupling function follows the same architecture: five hidden layers of 1,000 units for MNIST, and four layers of 2,000 units for SVHN and CIFAR-10.

The NICE models are trained with Adam with learning rate $10^{-3}$, momentum 0.9, $\beta_2 = 0.01$, $\lambda = 1$ and $\epsilon = 10^{-4}$. Our VPT models are trained with Adam with learning rate $0.5$.

**Computational complexity analysis.** Our complexity analysis shows that the computational cost of VPT scales primarily with the number of nodes per dimension, which is $O(2^{LD})$ for a tree of level $L$ and the feature dimension of $D$. In our implementation, we mitigate this cost by choosing a small value of $L$ (e.g., 2–4) and leveraging the independence across dimensions to parallelize computation. The recursive structure does introduce overhead due to tree traversal and node splitting, while these operations can be optimized via parallel processing across dimensions and nodes.

Table A2: Empirical profiling: training time per epoch for different datasets and configurations.

| Datasets | No. Samples | $D$ | Model | Tree Level ($L$) | Train Time/Epoch |
|---|---|---|---|---|---|
| GAS | $1,052,065$ | 8 | Block-NAF | – | 1 min 30 sec |
| | | | VPT | 4 | 2 min 10 sec |
| | | | VPT | 6 | 2 min 20 sec |
| POWER | $2,049,280$ | 6 | Block-NAF | – | 3 min |
| | | | VPT | 4 | 3 min 50 sec |
| | | | VPT | 6 | 4 min |
| MNIST | $60,000$ | 784 | NICE | – | 30 sec |
| | | | VPT | 2 | 35 sec |
| | | | VPT | 4 | 50 sec |
| CIFAR10 | $60,000$ | 1024 | NICE | – | 50 sec |
| | | | VPT | 2 | 1 min |
| | | | VPT | 4 | 1 min 20 sec |

Empirical profiling indicates that, when implemented on modern hardware, the additional overhead from VPT is modest compared to that of the normalizing flow backbone. Further optimizations—such as batch processing of tree nodes, caching intermediate results, and efficient tree traversal algorithms—can reduce computational burden even more. Future work will include a detailed benchmark over various $D$ and $L$ settings to quantify these improvements.

Furthermore, the additional memory footprint introduced by VPT is negligible, as shown in Table A3. The memory overhead remains below $0.05\%$, demonstrating that memory usage stays essentially constant when using VPT.

Table A3: Peak GPU memory usage (in gigabytes) during training with Adam and a batch size of 128 using Block-NAF as the backbone. Compared to vanilla Block-NAF with a Gaussian prior, VPTs do not introduce any additional memory overhead.

| Datasets | Gaussian Prior | VPT ($L = 4$) | VPT ($L = 6$) |
|---|---|---|---|
| POWER | 0.22 | 0.22 | 0.22 |
| GAS | 0.22 | 0.22 | 0.22 |
| HEPMASS | 5.00 | 5.00 | 5.00 |
| MINIBOONE | 4.04 | 4.04 | 4.04 |
| BSD300 | 19.83 | 19.83 | 19.84 |

It is important to note that the dimensionality $D$ referenced in the complexity analysis $\mathcal{O}(2^L D)$ typically corresponds to the latent representation rather than raw pixel dimensions. For image datasets, standard dimensionality reduction techniques such as convolutional layers effectively reduce $D$, making it practically manageable even for high-dimensional inputs.

## C  Additional Experiment Results on Density Estimation

**Density estimation and image generation on the SVHN dataset.**

In addition to the experiments in Section 4.3, we also conduct experiments on the SVHN dataset.

Figure A1 presents the density estimation results evaluated on the test set, while randomly generated images are shown in Figure A2. Similar to our findings with MNIST and CIFAR-10, our VPT enhances density estimation on SVHN without sacrificing perceptual quality.

**MNIST interpolation**. Figure A3 presents more results on the MNIST interpolations. Similar to Figure 4, we can observe that the interpolation images with VPT are more clear and meaningful than those with the traditional Gaussian and logistic prior distributions.

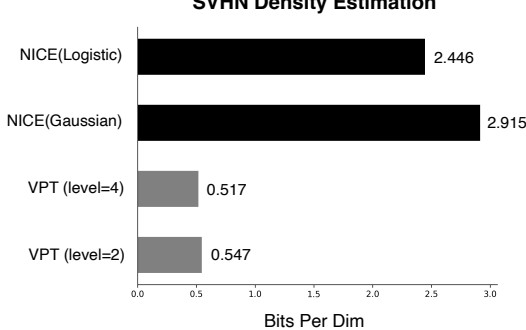

Figure A1: Negative log-likelihood in bits-per-dimension on the SVHN test set.

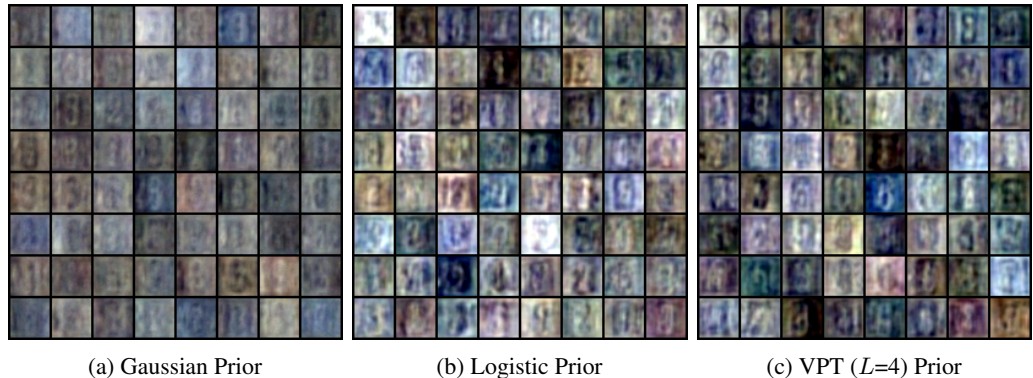

| (a) Gaussian Prior | (b) Logistic Prior | (c) VPT ($L$=4) Prior |

Figure A2: Randomly generated SVHN images with the NICE backbone under different priors.

**Perceptual quality**. Although perceptual quality is not our primary focus, Figure A4 shows that randomly sampling from our VPT prior produces images with clarity comparable to standard Gaussian or logistic priors. This indicates that VPT improves log-likelihood without sacrificing image generation quality, providing an advantageous balance between interpretability, uncertainty quantification, and generation performance.

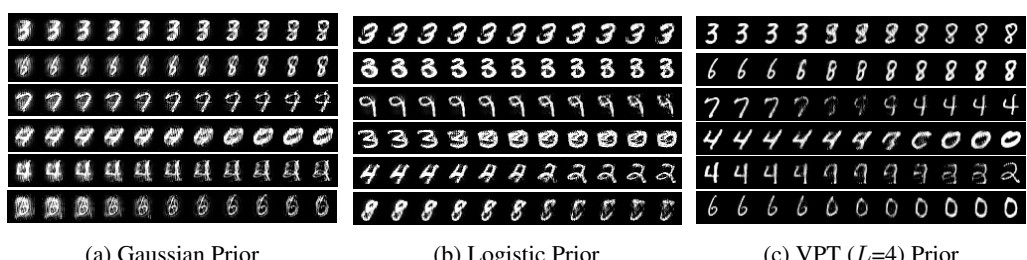

| (a) Gaussian Prior | (b) Logistic Prior | (c) VPT ($L$=4) Prior |

Figure A3: Interpolation on the MNIST images with the NICE backbone under different priors.

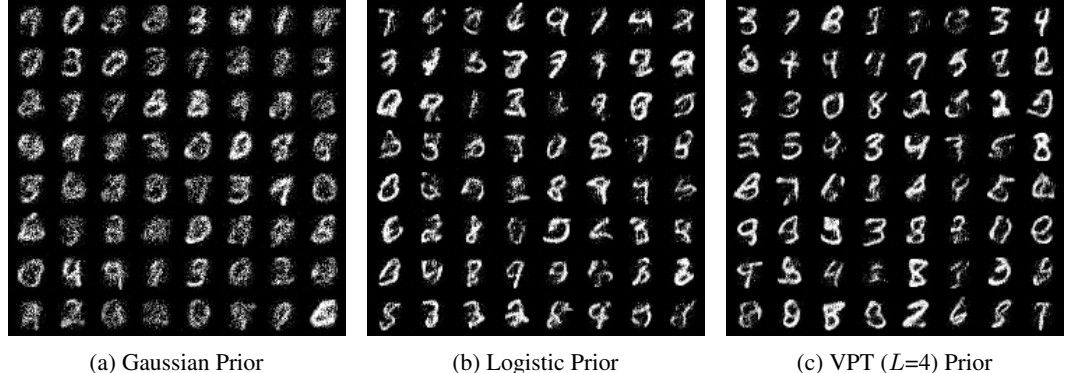

(a) Gaussian Prior         (b) Logistic Prior         (c) VPT ($L$=4) Prior

Figure A4: Randomly generated MNIST images with the NICE backbone under different priors.

