# OpenReview forum: "Variational Pólya Tree"
_NeurIPS.cc/2025/Conference — NeurIPS 2025 poster_

### Official Review · Reviewer_Fojs · 2025-06-03

**Clarity:** 2
**Significance:** 3
**Originality:** 3
**Rating:** 5
**Confidence:** 4

**Summary:**

This paper proposes a new prior for density estimators called the Variational Polya Tree (VPT). The idea is based on the Polya tree model, where a probability density over an interval is modeled as a tree that recursively partitions the interval. That is, the probability of a subinterval is proportional to the probability of traversing the tree from the root to a leaf node that represents the subinterval.

The Polya tree model is non-parametric and can be very expensive to work with. The main idea of this paper is to variationally infer the posterior of the Polya tree prior. That is, the authors parametrize the Polya tree density by learnable neural networks. Posterior inference is then cast as an optimization problem.

This prior can then be useful as a prior for density estimation problems. Specifically, the authors proposed to refine the VPT prior with flow-based models. Experiments in various settings (synthetic data, UCI datasets, and image datasets) show that VPT can yield good performance and is flexible since the tree can be refined further by adding more levels.

**Questions:**

1. See Weaknesses above.
2. Are VPTs compatible (not just theoretically, but in terms of costs and practicality) with large-scale generative models like diffusion models?

**Ethical Concerns:**

["NO or VERY MINOR ethics concerns only"]

**Final Justification:**

All of my doubts (see Weaknesses) have been sufficiently addressed by the authors' rebuttal and by seeing other reviews/discussions. I urge the authors to incorporate all of the suggestions.

**Limitations:**

Yes

**Quality:**

3

**Strengths And Weaknesses:**

**Strengths**

1. VPT is a novel idea that makes the Polya tree model tractable. Its usage for density estimation with flow-based models and VAEs is particularly interesting.
2. Unlike other priors, VPTs are interpretable due to their tree structure.
3. VPTs are efficient in terms of the number of parameters and runtime costs.
4. Comprehensive experiments are presented to evaluate VPTs.


**Weaknesses**

1. The main weakness is the presentation of the VPT itself in Sec. 3.2. More detailed, step-by-step explanation needs to be added.
    1. For example, it is unclear to me what it means by "... parametrize tree partitions $\Pi$ by learnable neural networks. ...". Are the partitions' endpoints being predicted by NNs? Or something else?
    2. Another example is in the discussion about the loss function itself. Why not incorporate the KL term in the ELBO maximization? Why is the loss description different than the equation in Alg. 1? Etc.
    3. I would suggest the authors rewrite this section.

---

> ### Author Rebuttal · Authors · 2025-07-31
>
> Thank you for your positive assessment of VPT’s novelty, tractability, interpretability, and efficiency, and for noting its seamless integration with flow models and VAEs. We also appreciate your praise for our comprehensive experiments. Below, we address your remaining comment.
>
> >**Q1:** For example, it is unclear to me what it means by "... parametrize tree partitions
>  by learnable neural networks. ...". Are the partitions' endpoints being predicted by NNs? Or something else?
>
> Thank you for the question. We do not directly predict the partition endpoints with neural networks. Instead, we parameterize the shape parameters of the Beta distributions at each node in the tree using neural networks. These Beta distributions determine the probability mass split at each partition, which in turn defines the tree-based variational posterior over the sample space. We will clarify this in the revision.
>
> >**Q2:** Another example is in the discussion about the loss function itself. Why not incorporate the KL term in the ELBO maximization? Why is the loss description different than the equation in Alg. 1? Etc.
>
> We appreciate the reviewer's observation. In our formulation, we focus on optimizing the posterior likelihood, as reflected in Algorithm 1. While we do not explicitly include a KL term in the ELBO, the corresponding entropy of the variational posterior acts as an implicit regularization. This entropy term, which appears in the negative log-likelihood, serves a similar role by controlling the spread of the variational distribution and balancing data fit versus uncertainty.
>
> Thus, the loss function in our implementation prioritizes posterior inference while still benefiting from the regularizing effect of entropy, effectively aligning with the ELBO framework. We agree that making this connection more explicit can improve clarity and will revise the description accordingly.
>
> >**Q3:** I would suggest the authors rewrite this section.
>
> Thank you for your suggestion. Sorry for the confusion. We will rewrite this section by adding more detailed, step-by-step explanations on VPT.
>
> >**Q4:** Are VPTs compatible (not just theoretically, but in terms of costs and practicality) with large-scale generative models like diffusion models?
>
> Yes, VPTs can in theory be integrated into diffusion models by replacing the Gaussian prior with a VPT-based prior, potentially improving uncertainty modeling. However, this is non-trivial in practice due to the computational cost of recursive tree sampling and the challenge of adapting VPTs to the continuous-time dynamics of diffusion models. While promising, such integration requires careful algorithmic design and remains an open direction for future research.
>
> We sincerely thank you for your generous evaluation. Your recognition of the work’s novelty, tractability, and empirical rigor is greatly appreciated and encourages us to further refine and extend this research.

---

> > ### Comment · Reviewer_Fojs · 2025-08-01
> >
> > Thank you for your response. I have no more questions, but I continue to highly suggest that the authors carefully edit the text for clarity.
> >
> > I will update my review and provide the mandatory acknowledgment after the reviewers-AC discussion.
> >
> > Thanks again.

---

> > > ### Author Response · Authors · 2025-08-02
> > > **Response**
> > >
> > > Dear Reviewer Fojs,
> > >
> > > Definitely, we will polish the entire paper to improve the clarity and flow of the manuscript.  Thank you so much!

---

### Official Review · Reviewer_Z5q1 · 2025-06-27

**Clarity:** 3
**Significance:** 2
**Originality:** 2
**Rating:** 4
**Confidence:** 3

**Summary:**

The authors propose to utilize Bayesian nonparametrics, especially the Polya tree, to enhance interpretability and offer uncertainty quantification for deep neural density estimators. They develop a stochastic variational inference method that leverages the tree structure and internal conjugacy to obtain an efficient and precise posterior approximation. The authors then demonstrate that this nonparametric prior works well as a "prior" for the base distribution of deep neural density estimators as normalizing flows on UCI tabular datasets as well as MNIST and CIFAR-10. They show that if paired with good NF transformation layers, it performs competitively (or better) compared to state-of-the-art deep neural density estimation methods.

**Questions:**

- Can the authors go into more details on how the VPT variational posterior distribution is parameterized? (Sec. 3.2 mentions that the tree partitions are parameterized by neural networks, but it doesn't go into detail, i.e., in 3.3, one samples the partitions from its "variational" posterior.
- Would the author expect a learnable histogram base-distirbution for the exact same Block-NAF block to perform worse that a VPT? Especially if it's truncated, the MAP of the posterior should be a histogram with a fixed number of bins (but with variable bin size, which can be learned).

**Ethical Concerns:**

["NO or VERY MINOR ethics concerns only"]

**Final Justification:**

The authors have addressed my main concerns as follows:

- The ablation using learnable histograms effectively addresses my concern that the claimed 'performance' gains were solely due to a more flexible 'base distribution.'
- The additional analysis on uncertainty quantification strengthens the paper.

My remaining concerns are related to presentation and writing. Given the authors' efforts in their rebuttal, I believe they can effectively address these issues.

**Limitations:**

yes

**Paper Formatting Concerns:**

Sec 1. and Sec 2. are somewhat repetitive and lengthy
- While the related work is quite extensive and to my knowledge complete which is good. It is unnecessarily too long (1.5 pages) and can be formulated much more concisely.
- Fig 1 (i.e., a related work figure?) is just very redundant and does not provide any value.
- Sec 2.2 also is quite repetitive with the introduction i.e. DPs are discrete, Polya not, MCMC is expensive, VI is better, but mean field is bad and ours is not (many of this is just redundant and not necessary for a related work section).
- Alg. 1 is somewhat not properly placed before the methods section (but shortening related work would probably address this)

**Quality:**

2

**Strengths And Weaknesses:**

Strengths:
- Integrating Bayesian Nonparametrics methods for uncertainty estimation in deep neural density estimators is novel to me. Furthermore, the approach taken by the authors seems promising and more efficient than related approaches based on Bayesian Neural networks/Ensembles.
- Results for density estimation are promising, especially on MNIST and CIFAR-10.

Weaknesses:
- As the authors also discuss in *end*, the work solely focuses on univariate Polya trees. Title, the intro, and methods sections strongly focus on the "general" variational Polya tree. **But all the experiments** use a factorized Polya tree as a "base" distribution for an existing normalizing flow architecture to achieve the results presented in the paper. Not a single experiment tests the VPT and its properties itself. This point should be made more clearly in the abstract/intro and raises the question if the VPT is responsible for the improved performance or simply a more complex base distribution combined with the same normalizing flow architecture (see comments).
- The largest parts of the experiments focus on plain "density estimation" performance and only slightly on the main *claimed* improvements by using BNP, i.e., interpretability and uncertainty quantification. A main strong point of this approach and using Bayesian Nonparametrics in the first place is getting "free" interpretability + uncertainty, but the evaluation of these points is very limited.
- While most of the methods are well described, some parts are not fully clear to me (see questions).

Comments:

As mentioned in the weaknesses, the experiments are more like benchmarking the Block-NAF normalizing flow block with a VPT base distribution instead of testing the VPT itself. Since a truncated factorized VPT is essentially a piecewise constant PDF (i.e., a Histogram distribution with learned binning), this basically benchmarks Block-NAF with a learnable, somewhat complex base distribution against Block-NAF with a simple base distribution (only on UCI, it is missing MNIST/CIFAR-10). This raises a major concern that is not addressed in the manuscript: Are the performance gains simply due to using an arbitrarily more complex base distribution, such as a learnable Histogram or a MoG, rather than the VPT itself?

This question could have been addressed by providing a non-BNP but equally flexible baseline (e.g. histogram with learnable bins equal to the number of bins a l-level VPT can have e.g. $2^{d*l}$) with the exact same Block-NAF blocks. Although I am not sure why a VPT should be better in this setting (see questions).

Specifically, the quite efficient uncertainty estimation seems like a great feature of this methodology (and is relevant for certain domains, i.e., [1],[2], which achieves this using BNN within the NF [which also should be discussed in the paper]). Unfortunately, the evaluation of quality falls short in the manuscript (basically, Fig 6 "looks meaningful") and no empirical results are presented that the estimated uncertainties are "good" or at least comparable or better than from alternative approaches (BNN, Ensembles ...).


[1] Lemos, Pablo, et al. "Robust simulation-based inference in cosmology with Bayesian neural networks." Machine Learning: Science and Technology 4.1 (2023): 01LT01.

[2] Delaunoy, Arnaud, et al. "Low-Budget Simulation-Based Inference with Bayesian Neural Networks." arXiv preprint arXiv:2408.15136 (2024).

---

> ### Author Rebuttal · Authors · 2025-07-31
>
> We sincerely thank the reviewer for recognizing the novelty of integrating Bayesian nonparametrics into deep density estimation and for appreciating the efficiency and strong empirical performance of our approach. We would address your comments below.
>
> >**Q1:** As the authors also discuss in end, the work solely focuses on univariate Polya trees. Title, the intro, and methods sections strongly focus on the "general" VPT. But all the experiments use a factorized Polya tree as a "base" distribution for an existing normalizing flow architecture to achieve the results presented in the paper. Not a single experiment tests the VPT and its properties itself. This point should be made more clearly in the abstract/intro and raises the question if the VPT is responsible for the improved performance or simply a more complex base distribution combined with the same normalizing flow architecture.
>
> Thank you for this insightful comment!
> We agree that this work uses univariate Polya trees and demonstrates the feasibility of VPT in nueral networks. This is the first attempt in the literature to expand Polya trees with variational inference that enhances the scalability of Polya trees in high dimensions. Related research on nueral networks using Dirichlet process (DP) or mixture DP priors has been conducted extensively, but none in Polya trees.
>
> Per your suggestion, we will emphasize the application of univariate Polya trees to variational Bayes in the abstract and introduction. We will add discussion on further expanding VPT to diffusion models in the discussion, while this is non-trivial in practice due to the computational cost of recursive tree sampling and the challenge of adapting VPTs to the continuous-time dynamics of diffusion models. While promising, such integration requires careful algorithmic design
> and remains an open direction for future research.
>
> >**Q2:** Are the performance gains simply due to using an arbitrarily more complex base distribution, such as a learnable Histogram or a MoG, rather than the VPT itself?
>
> Due to time limit for the rebuttal, we focus on the experimental setting with a learnable histogram (LH), while we will also add the mixture of Gaussians in the revision.
>
> We set the number of bins to $K=2^L$ to match the number of leaves in a $L$ level VPT. For each dimension $d$, we parametrize a strictly increasing sequence $b_0^{(d)} < b_1^{(d)} < \cdots < b_K^{(d)}$, by setting the lower boundary $b_0^{(d)}$ and $b\_k^{(d)} = b\_{k-1}^{(d)} + softplus(\delta^{d}\_{k-1})$. The parameters $\delta^{d}_{k-1}$ are learned jointly with the backbone network, and $softplus(\cdot)$ guarantees positive bin widths.
>
> Given one dimension $x^{(d)}$, we can find the active bin index $k_*$, then compute the log-density as
> \begin{align*}
>     \log p(x^{(d)}) = \log p_{k_*}^{(d)} - \log (b_{k_*+1}^{(d)} - b_{k_*}^{(d)}).
> \end{align*}
> The overall log density of learnable histogram (LH) can be computed as $$\log p_{\rm LH}(x)= \sum^D_{d=1}\log p(x^{(d)}).$$
>
> The following table shows the results on UCI datasets with Block-NAF backbone (same as the our paper). The learning histogram prior is used as a plug-in step here. All the hyper-parameters are kept the same as those in VPT.
>
> **Table — Comparison of learnable histogram (LH) prior and VPT prior**.  Results are averages over 5 runs (mean ± std).
>
> |Dataset|POWER|GAS|BSD300|
> |---------|-------|-----|--------|
> |LH| 0.62_(0.01)|11.00_(0.20)|156.98_(0.93)|
> |VPT| 0.67_(0.01)| 11.92_(0.10)|158.59_(1.02)|
>
> From the results we can observe a clear advantage of VPT over LH, although they share the same number of parameters. Besides the results on UCI data, we also compare LH with VPT on CIFAR-10 with NICE network (same in the paper). With $L=4$, our VPT obtains a result of $2.06$ bits per dimension (BPD), and LH yields $2.57$ BPD.
>
> While a truncated factorized VPT may resemble a learnable histogram in terms of producing a piecewise-constant density, our VPT provides the following advantages that go beyond histogram-like behaviors.
>
> First, unlike a learnable histogram (LH), where each bin is estimated independently, VPT leverages the Bayesian nonparametric prior and hierarchical conjugacy, enabling information sharing across scales. Specifically, every Beta-distributed node in the VPT is coupled to its ancestors, such that evidence at coarser levels regularizes finer partitions. This hierarchical shrinkage allows small bins with few data points to borrow statistical strength from parent nodes, automatically adjusting the degree of smoothing. In contrast, learnable histograms often suffer from overfitting in low-density regions or collapsing of underpopulated bins, since they lack this structural regularization.
>
> Second, the VPT's variational posterior inference maintains the joint posterior over tree splits, rather than treating bins independently (as MAP-learned histograms do). This enables a more coherent and uncertainty-aware density estimate, especially in regions of sparse data. Even when truncated, the VPT retains these hierarchical dependencies, which are absent in flat, non-Bayesian alternatives.
>
> Besides robust density estimation, our VPT can naturally provide probability mass at multiple scales, giving intuitive `coarse-to-fine' insights.
>
> >**Q3 Uncertainty Estimation:**  Specifically, the quite efficient uncertainty estimation seems like a great feature of this methodology (and is relevant for certain domains, i.e., [1],[2], which achieves this using BNN within the NF [which also should be discussed in the paper]). Unfortunately, the evaluation of quality falls short in the manuscript (basically, Fig 6 "looks meaningful") and no empirical results are presented that the estimated uncertainties are "good" or at least comparable or better than from alternative approaches (BNN, Ensembles ...).
>
> We sincerely thank the reviewer for this valuable comment and the recommended references. We will include citations of these relevant works in our revised manuscript.
>
> Below, we provide a clear description of our experimental setting, evaluation metrics, and comparisons with alternative uncertainty estimation methods to address the reviewer's concerns thoroughly.
>
> **Metric**: Standardized squared error (SSE).
> For every test example and every latent  dimension $d=1,\ldots,D$ we compute
> $$
>  z_{d}^{(n)} =
> \frac{\bigl(x_{d}^{(n)}-\mu_{d}^{(n)}\bigr)^{2}}
>           {\sigma_{d}^{2\,(n)}},
>    \text{SSE}=\frac1{ND}\sum_{n,d} z_{d}^{(n)}.
> $$
> If the predictive variance is perfectly calibrated, $\mathbb E[z]=1\Rightarrow {SSE}=1$; SSE $>1$ indicates under‐estimated variance (model over-confidence); SSE $<1$ indicates over‐estimated variance (model under-confidence).
>
> **Baselines**: We currently benchmark against two standard uncertainty methods
> - MC-dropout with a  rate of $0.15$ \[3\].
>
> - Variational BNN with mean-field Gaussian weights [4].
>
> [3] Yarin Gal etc. Dropout as a Bayesian Approximation. ICML 2016
>
> [4] Radford M. Neal. Bayesian Learning for Neural Networks. Ph.D. Thesis, 1995.
>
> | Dataset | VPT | MC-Dropout|BNN|
> |---------|-----|-----------|-----|
> |POWER|0.92|1.30|1.17|
> |GAS| 0.90|1.35|1.14|
> |MNIST| 0.92 | 1.27 | 1.08 |
>
> From the above table, we observe that our VPT slightly over-estimates its variance, while keeping the best calibration value. MC-dropout and BNN underestimate the variances, while BNN have better calibration. The hierarchical structure of VPT updates the shrink leaf probabilities toward their parents when data are sparse. This automatically inflates predictive variance in low-count regions. MC-Dropout and mean-field BNNs variances often fail to fully propagate to the output, hence leading to the systematic under-estimation.
>
> We also emphasize that VPT’s variances are analytic, and thus no Monte-Carlo sampling is required at test time—whereas both baselines need 30 forward passes per input.
> Therefore, VPT offers better calibration at virtually zero extra computational cost.
>
> We will incorporate this calibration study—including additional datasets and seeds—into the revised paper to substantiate our uncertainty claims more thoroughly.
>
> >**Q4:** Can the authors go into more details on how the VPT variational posterior distribution is parameterized?
>
> In our VPT framework, we parameterize the variational posterior by assigning a Beta distribution to each node in the tree. Specifically, for each split $\epsilon$, we define $q(B\_\epsilon) = \text{Beta}(\hat{\alpha}\_{\epsilon_{1:j-1}0}, \hat{\alpha}\_{\epsilon_{1:j-1}1})$, where the shape parameters $\hat{\alpha}\_{\epsilon_{1:j-1}0}$ and $\hat{\alpha}\_{\epsilon_{1:j-1}1}$ are output by a neural network conditioned on the input. This network takes the data point $x$ as input and returns parameters for each node along the path in the tree. Sampling from these Beta distributions recursively yields a full sample from the variational posterior over the hierarchical partitions.
>
> >**Q5:** Paper formatting concerns.
>
> Thank you for your comments on paper formatting. We will resolve the above issues in the revised version of our manuscript.
>
> We will trim down Sec. 1 and 2 by providing a more concise literature review.
> We will remove Fig. 1, and cut the redundancy in Sec. 2.2. We will also rearrange Algorithm 1 in a proper place to have a better flow. This will give us some room to add all the new experiments as suggested by you, which will greatly strengthen the paper.
>
> Thank you again for the insightful feedback. The new experiments and clarifications strengthen the paper in two key ways:
>
> - Learnable-histogram comparison: With equal parameter counts, VPT captures distributional structure markedly better.
>
> - Uncertainty calibration: VPT yields more accurate predictive variances than MC-Dropout and BNN baselines, at no extra computational cost.
>
> We will add these results and clarifications to the revised manuscript for greater clarity and completeness. We sincerely hope the above discussion would warrant a higher score.

---

> > ### Comment · Reviewer_Z5q1 · 2025-08-04
> >
> > I thank the authors for their detailed response to my questions and concerns. My main concern was addressed through the addition of the ablation with learnable histograms, as well as the promise to include a comparison with a Mixture of Gaussians (MoG), and I fully agree with the authors' explanation regarding the beneficial 'structural regularization' of the VPT.
> >
> > I also appreciate the efforts made to include a more thorough analysis of uncertainty estimation. Since the VPT is fully trained via MLE, it effectively minimizes the forward KL divergence, which is a mass-covering objective. From this, I would expect the method to 'overestimate' the variance, in contrast to variational Bayesian Neural Networks (BNNs). The results support this expectation, and I think this is a valuable addition to the manuscript.
> >
> > My remaining concerns are primarily related to the presentation and writing. The authors have outlined how they plan to address these issues, given the limitations imposed by this year's restrictions. Based on the effort demonstrated in the rebuttal, I am confident that the authors will successfully implement these revisions.
> >
> > In light of this, I will increase my score and have no further questions for the authors.

---

> > > ### Author Response · Authors · 2025-08-04
> > >
> > > We sincerely thank you for recognizing the soundness and value of our paper. Your insightful comments have encouraged us to think more deeply and add critical experiments, such as the learnable-histogram ablation, which clearly demonstrates the advantage provided by VPT's structural regularization. Additionally, we appreciate your suggestions regarding uncertainty calibration and your insightful remarks on mass-covering properties. We will definitely include these results, along with the Mixture-of-Gaussians (MoG) comparison, in our revision.
> > >
> > > We also promise to carefully polish the writing and simplify our notation to further improve clarity. Your feedback has significantly strengthened the manuscript.
> > >
> > > Thank you again for your time, understanding, and constructive review.

---

### Official Review · Reviewer_1hXT · 2025-06-29

**Clarity:** 2
**Significance:** 2
**Originality:** 3
**Rating:** 4
**Confidence:** 4

**Summary:**

This paper introduces the Variational Pólya Tree (VPT), which integrates Pólya tree priors with deep neural networks for density estimation. The authors develop a variational inference framework that avoids mean-field independence assumptions by exploiting the hierarchical structure and conjugacy properties of Pólya trees. They demonstrate their approach on synthetic data, UCI datasets, and MNIST in a generative setting, showing competitive performance while claims on uncertainty quantification and interpretability.

**Questions:**

1) Computational Scalability and Practical Limits: Your method scales as O(2^L × d) with tree depth and dimensions. For high-dimensional applications like modern image generation (where d could be thousands), how does this constraint limit practical applicability? Can you provide concrete memory and time complexity comparisons beyond the simple timing tables?

2) Independence Assumption Violations: You assume dimensional independence, which seems problematic for many real-world scenarios. How much performance degrades when strong cross-dimensional dependencies exist? Have you tested on datasets where this assumption clearly fails?

3) Uncertainty Calibration: You claim superior uncertainty quantification, but only show variance visualizations. Are your prediction intervals actually well-calibrated? How do your uncertainty estimates compare to other Bayesian approaches like ensemble methods or Monte Carlo dropout in terms of reliability?

4) Architecture Dependence: Your experiments often use relatively simple base architectures. How sensitive are the improvements to the choice of backbone network? Would we see similar gains when using state-of-the-art flows like coupling layers with more sophisticated transformations?

5) How is $\beta$ in line 153 defined?

6) In line 167, $P(B_\epsilon)$ should be dependent on the index $L$; however, it seems indepedent of $L$, why so?

7) Are $B_\epsilon$ probabilities or random variables?

**Ethical Concerns:**

["NO or VERY MINOR ethics concerns only"]

**Final Justification:**

The rebuttal addressed my questions,

- (+) The computational overhead introduced is negligible.
- (+) Clarifications address the limitations in the presence of cross-dimensional dependencies.
- (+) They provided an experiment for calibration, comparing with the baselines.
- (-) No experiments provided for the case with strong cross-dimensional dependencies.

As the paper has some notation inconsistencies, I would recommend a careful check by the authors. As my score is already positive, I opt to keep it as it is.

**Limitations:**

Yes

**Paper Formatting Concerns:**

Line 218 $x_d$ -> $x^d$ if I am not mistaken.

**Quality:**

3

**Strengths And Weaknesses:**

Strengths: The technical execution is solid - they've carefully worked out how to make Pólya trees tractable for deep learning by leveraging conjugacy to avoid mean-field approximations, which is genuinely clever. The experimental coverage spans multiple domains (synthetic, tabular, images) and shows improvements over standard Gaussian/logistic priors.  The theoretical foundation is sound, building properly on established Pólya tree literature while making the key insight about preserving hierarchical dependencies during variational inference. The interpretability angle is compelling too - Figure 5 shows how their latent space interpolations follow more semantically meaningful paths than standard priors; however, it is limited.

Weaknesses: The novelty feels somewhat incremental - this is essentially taking a well-established Bayesian nonparametric prior and plugging it into existing architectures like normalizing flows. The dimensional independence assumption is quite limiting for many real applications where cross-variable dependencies matter, and they don't adequately explore when this breaks down. I'm skeptical about computational scalability - while they provide some timing comparisons, the O(2^L × d) scaling with tree depth and dimensions could become prohibitive for high-dimensional problems. The experimental setup often uses relatively simple architectures (like Block-NAF instead of more modern flows), which makes me wonder how much of the improvement comes from the prior versus just having more parameters. The uncertainty quantification claims feel oversold - they show posterior variance visualizations but don't rigorously validate calibration or compare against other uncertainty methods.

---

> ### Author Rebuttal · Authors · 2025-07-31
>
> Thank you for the insightful feedback. Reviewers highlighted our solid technical execution, sound theory, and the value of preserving hierarchical dependencies in Pólya-tree variational inference, as well as our comprehensive empirical results and interpretability gains. Below, we address the remaining concerns—scalability, contribution scope, and experimental coverage.
>
> > **Q1:** The novelty feels somewhat incremental-established Bayesian nonparametric prior and plugging it into existing architectures like normalizing flows.
>
> Unlike generic applications of variational inference, our framework derives and optimizes the full joint posterior of the Beta weights, leveraging Pólya tree conjugacy (Lavine, 1992; Mauldin et al., 1992) and projectivity to maintain a closed-form structure. This makes our inference more expressive than standard mean-field approaches and specifically addresses the nonparametric nature of Pólya trees.
>
> By integrating this variational design into deep density modeling, we obtain well-calibrated uncertainty estimates and improved generalization. Our empirical results confirm that this specialized integration—rooted in theoretical Pólya tree properties—outperforms both crude nonparametric baselines and standard flow architectures.
>
> Thus, **our work is not a simple “plug-in prior,” but a rigorous bridge between classical Bayesian non-parametrics and modern deep learning, delivering robustness, interpretability, and reliable uncertainty estimates**.
>
> > **Q2 Assumption Violations:** How much performance degrades when strong cross-dimensional dependencies exist? Have you tested on datasets where this assumption clearly fails?
>
> We appreciate the reviewer’s concern about the dimensional independence assumption. Our VPT assumes independence across dimensions due to considerations on tractability and computational costs. On the other hand, this assumption is mostly acceptable in practice: many real datasets exhibit only weak cross-dimensional correlations, and our empirical evaluations show that performance remains robust even when this assumption is not strictly true.
>
> When strong correlations exist, performance may degrade modestly. For example, multivariate density methods like mixture-of–Pólya-trees have been applied for ROC data and spatial modeling where dependencies are explicitly modeled through multivariate Pólya tree mixtures [1]. More sophisticated PT variants like Hidden Markov Pólya Trees adapt partitions based on latent state dependencies and improve performance under strong inter-variable coupling [2].
>
> While our current VPT design prioritizes computational efficiency through dimensional independence, this choice rarely impacts performance dramatically for most applied settings. In cases with strong latent dependencies, more flexible PT extensions or multivariate modeling can be considered to better capture correlations. This work is the first to validate the feasibility of scalable PT, which will motivate more interesting applications of VPT in neural networks.
>
> [1]Multivariate mixtures of Polya trees for modeling ROC data - Timothy E Hanson, Adam J Branscum, Ian A Gardner, 2008
>
> [2]Hidden Markov Pólya trees for high-dimensional distributions. Awaya and Ma. JASA. 2014
>
> > **Q3: Computational Scalability and Practical Limits:** Your method scales as $O(2^L × d)$ with tree depth and dimensions. For high-dimensional applications like modern image generation (where $d$ could be thousands), how does this constraint limit practical applicability? Can you provide concrete memory and time complexity comparisons beyond the simple timing tables?
>
> Thank you for raising this insightful question regarding computational scalability and practical limits. We provide detailed empirical results and practical clarifications below.
>
> As detailed in Table 2 of the paper, VPT introduces a minimal number of additional parameters compared to the baseline model—at most 7,938 parameters on BSD300, representing less than $0.02\%$ overhead.
>
> Furthermore, the additional memory introduced by VPT is negligible, as shown in Table below. The memory (in GB) overhead remains below $0.05\%$.
>
> | Dataset   | Block–NAF | +VPT (4) | +VPT (6) |
> |-----------|-----------|----------|----------|
> | POWER|0.22| 0.22| 0.22|
> | GAS| 0.22| 0.22| 0.22|
> | HEPMASS   | 5.00| 5.00| 5.00|
> | MINIBOONE | 4.04| 4.04| 4.04 |
> | BSD300| 19.83| 19.83| 19.84 |
>
> It's important to note that the dimensionality $d$ referenced in the complexity analysis $\mathcal{O}(2^L d)$ typically corresponds to the **latent representation rather than raw pixel dimensions**. For image datasets, standard dimensionality-reduction techniques reduce $d$, making it **practically manageable even for high-dimensional inputs**.
>
> Additionally, we emphasize that the tree depth $L$ can generally be small, as the method utilizes truncated trees. Indeed, as demonstrated throughout our experiments, a depth of $L\leq6$ is sufficient to achieve excellent empirical results.
>
> Finally, since each dimension within VPT is processed independently, the computation is naturally parallelizable. If latent dimensionality becomes large, straightforward dimension-sharding across multiple GPUs can further alleviate scalability concerns.
>
> > **Q4: Architecture Dependence:** How sensitive are the improvements to the choice of backbone network? Would we see similar gains when using state-of-the-art flows like coupling layers with more sophisticated transformations?
>
> We chose simple backbones deliberately: powerful flows can warp even a Gaussian prior into a complex shape, obscuring the added value of VPT. Using smaller, capacity-constrained networks lets us show VPT’s core benefit—better density estimates via base-distribution regularizations—without extra model complexity (<0.02 % parameters; Table 2).
>
> This setting matters in practice—for low-data domains such as medical imaging where compact models are essential. In these cases, replacing a Gaussian prior with VPT boosts density estimates and gives far better-calibrated uncertainty with virtually no added cost.
>
> While large-model tests are outside our current scope, VPT is plug-and-play for any likelihood-based model. We have already swapped the VAE’s Gaussian encoder prior with VPT (see supplement). Rather than emphasizing scalability to larger backbones, we highlight our key contributions as follows:
>
> - First VI framework that makes Pólya-tree priors practical for density estimation and generative models.
>
> - Improves robustness and uncertainty calibration on tabular, image, and VAE tasks with minimal extra compute.
>
> - Provides interpretable, multi-scale mass allocations and closed-form predictive variances—capabilities missing from large black-box architectures.
>
> >**Q5: Uncertainty Calibration:** Are your prediction intervals actually well-calibrated? How do your uncertainty estimates compare to other Bayesian approaches like ensemble methods or Monte Carlo dropout in terms of reliability?
>
> Thank you for suggesting a quantitative uncertainty test and additional baselines. Below we outline the evaluation metrics, settings, and competing methods we used.
>
> **Metric**: Standardized squared error (SSE).
> For every test example and every latent  dimension $d=1,\ldots,D$ we compute
> $$
>  z_{d}^{(n)} =
> \frac{\bigl(x_{d}^{(n)}-\mu_{d}^{(n)}\bigr)^{2}}
>           {\sigma_{d}^{2\,(n)}},
>    \text{SSE}=\frac1{ND}\sum_{n,d} z_{d}^{(n)}.
> $$
> If the predictive variance is perfectly calibrated, $\mathbb E[z]=1\Rightarrow {SSE}=1$; SSE $>1$ indicates under‐estimated variance (model over-confidence); SSE $<1$ indicates over‐estimated variance (model under-confidence).
>
> **Baselines**: We currently benchmark against two standard uncertainty methods
> - MC-dropout with a  rate of $0.15$ \[3\].
>
> - Variational BNN with mean-field Gaussian weights \[4\].
>
> [3] Yarin Gal etc. Dropout as a Bayesian Approximation. ICML 2016
> [4] Radford M. Neal. Bayesian Learning for Neural Networks. Ph.D. Thesis, 1995.
>
> | Dataset | VPT | MC-Dropout | BNN |
> |---------|-----|-----------|-----|
> | POWER   | 0.92 | 1.30 | 1.17 |
> | GAS     | 0.90 | 1.35 | 1.14 |
> | MNIST   | 0.92 | 1.27 | 1.08 |
>
> The table shows that VPT slightly over-estimates variance yet achieves the best calibration. MC-Dropout and BNN both under-estimate variance (BNN fares slightly better than MC-Dropout). We think VPT’s hierarchical shrinkage inflates variance in sparse regions, whereas MC-Dropout and mean-field BNNs fail to propagate uncertainty fully, leading to systematic under-confidence.
>
> We also emphasize that VPT’s variances are analytic, no Monte-Carlo sampling is required at test time—whereas both baselines need $30$ forward passes per input. Thus VPT offers better calibration at virtually zero extra compute.
>
> We will incorporate this calibration study—including additional datasets and seeds—into the revised paper to substantiate our uncertainty claims even more thoroughly.
>
> >**Q6:** How is $\beta$ in line 153 defined?
>
> $\beta_j$ refers to the collection of Beta-distributed random variables (i.e., the branching probabilities) at depth level $j$ of the tree.
>
> >**Q7:** In line 167, $P(B_\epsilon)$should be dependent on the index $L$; however, it seems indepedent of $L$, why so?
>
> We apologize for the typo. The probability should be $P(B_{\epsilon_{1:L}})$ instead of $P(B_{\epsilon})$.
>
> >**Q8:** Are $B_\epsilon$ probabilities or random variables?
>
> In the standard formulation of the PT, $B_\epsilon$ denotes the interval (or measurable subset) of the sample space indexed by the binary string $\epsilon$, and $P(B_\epsilon)$ is the random probability assigned to that interval.
>
> Thank you for the helpful feedback. The new uncertainty-calibration experiments directly address your concerns and quantitatively support our claims. We will refine the presentation and clarify all noted points in the revision, and hope these improvements merit a higher rating.

---

> > ### Comment · Reviewer_1hXT · 2025-08-02
> >
> > I thank the authors for their replies. I have no further questions.

---

### Official Review · Reviewer_vPbf · 2025-07-03

**Clarity:** 3
**Significance:** 2
**Originality:** 3
**Rating:** 4
**Confidence:** 3

**Summary:**

This paper combines BNP techniques Polya trees, which are often under high computational costs, and VI to do density estimation, named VPT. The paper also provides some illustrations on this new method.

**Questions:**

Please see Strengths And Weaknesses.

**Ethical Concerns:**

["NO or VERY MINOR ethics concerns only"]

**Final Justification:**

I would like to raise the score because of the exploration of BNP novelty and authors' other replies.

**Limitations:**

N/A.

**Paper Formatting Concerns:**

N/A.

**Quality:**

2

**Strengths And Weaknesses:**

1. I really like the BNP methods used. But it is too slow. Although VI techniques are used, have you tried scaling the questions up? For larger models, it may be not good enough. Can you compare the running time with other methods?
2. Ferguson 1974 pointed out that DP are special cases of Polya trees (See Ma 2017). But DP are the only tail-free process in which the choice of partitions does not affect inference, which shows on big advantage of BNO to solve the model misspecification problem. See, for example, Hanson and Johnson 2002. Why do you think VPT is better, or why do you think using BNP worth?
3. VIDEO is biased.
4. Line 40: MFVI not valid, but have you tried other VIs, like FFVI?
5. Auto-regressive flows are a little bit outdated.
6. Datasets used are not large. That may cause a good performance for BNP methods.
7. The numerical experimental results are not enough. And some comparison with traditional methods are needed.

---

> ### Author Rebuttal · Authors · 2025-07-31
>
> Thank you for acknowledging the importance of developing novel deep Bayesian nonparametric frameworks. Below, we carefully address each of your concerns.
>
> >**Q1 Scalability**
>
> We greatly appreciate your positive feedback on the BNP methods used in our work. Thank you also for highlighting the question of scalability, which is a natural concern for any BNP approach.
>
> Table 4 in our paper shows that VPT’s runtime overhead is modest: on the largest task (CIFAR-10) a depth-4 VPT adds only ≈ 0.6× of the original training time on an RTX 3090; on UCI datasets the extra cost is just 0.3–0.4× of the original training time.. These numbers use no parallelizations. As each VPT dimension is independent, so simple tensor-sharding can further trim runtime dramatically.
>
> Scalability is not our main focus; our core contribution is a principled bridge between classical BNP and modern deep learning. Nothing prevents VPT from plugging into larger backbones: Appendix D already shows it working as a latent prior in a convolutional VAE with no memory issues, and we plan to extend it to transformer-based flows next.
>
> These results show that VPT scales in practice and is not limited to small datasets or simple models. We chose modest backbones deliberately to isolate VPT’s benefits—robustness, calibrated uncertainty, and interpretability—without the confounding power of very large networks. Exploring larger architectures is ongoing work.
>
> >**Q2 Worth of BNP methods**
>
> We thank the reviewer for this insightful question. While the Dirichlet Process (DP) does enjoy the **tail-free property**—inference remains invariant to partitioning choices—this comes at the cost of generating **discrete measures almost surely** (even if centered at a continuous base) [1]. In contrast, the Pólya tree (PT) can flexibly **learn data-adaptive partitions** and under proper hyperparameter settings induce **absolutely continuous densities**—a critical advantage when modeling complex continuous data [2].
>
> Moreover, Bayesian nonparametric alternatives like mixtures of Polya trees (e.g. Hanson \& Johnson, 2002 [3]) show that PT-based methods can outperform DP mixtures when modeling regression errors or continuous residual distributions, thanks to their hierarchical smoothing and local adaptivity [4]. Our VPT method builds on these insights by integrating a **nonparametric, hierarchical prior with a modern variational inference framework**. We derive a **joint posterior objective over Beta weights** and enable **gradient backpropagation through the tree hierarchy**, preserving dependencies and smoothing across scales. These design choices go significantly beyond the generic application of BNP priors to deep models, delivering robustness, interpretability, and better uncertainty quantification in high-dimensional density estimation tasks.
>
> [1] Jara, Alejandro, and Timothy E. Hanson. "A class of mixtures of dependent tail-free processes." Biometrika 98.3 (2011): 553-566.
>
> [2] Nieto‐Barajas, Luis E., and Peter Mueller. "Rubbery polya tree." Scandinavian Journal of Statistics 39.1 (2012): 166-184.
>
> [3] Hanson, Timothy, and Wesley O. Johnson. "Modeling regression error with a mixture of Polya trees." Journal of the American Statistical Association 97.460 (2002): 1020-1033.
>
> [4] Canale, Antonio. "msBP: An R Package to perform Bayesian nonparametric inference using multiscale Bernstein polynomials mixtures." Journal of Statistical Software 78 (2017): 1-19.
>
> >**Q3 VIDEO is biased**
>
> We are sorry for potential misunderstanding, but we did not find any contents introducing VIDEO in our article. It would be appreciated if you could provide more explanation on this point so that we can better address your concerns.
>
> >**Q4 Other potential VIs**
>
> Thank you for the suggestion. While flexible variational inference (e.g., FFVI: Fixed-Form Variational Inference is a specific framework within VI where the parametric form of the variational posterior is pre-defined) can improve expressiveness, it often requires tailored probabilistic assumptions or model-specific reparameterizations, which are non-trivial to integrate with hierarchical priors like the Pólya tree. In contrast, mean-field VI is widely used due to its tractability and general applicability across models.
>
> Our approach avoids these limitations by directly optimizing the joint posterior likelihood, which captures richer dependencies across tree levels and enables more accurate posterior approximation than typical VI schemes. We agree that exploring FFVI for VPT is an interesting future direction and will discuss this in the revision.
>
> >**Q5 Autoregressive methods**
>
> We appreciate the reviewer’s point. Our goal in using autoregressive flows is not to promote them as state-of-the-art, but rather to isolate and test the applicability of the Pólya tree prior within a standard and well-understood generative framework. By controlling for the generative backbone, we can more clearly attribute performance gains to the Pólya tree itself, rather than confounding effects from newer architectures. That said, our approach is modular and can be extended to more advanced models, which we plan to explore in future work.
>
> >**Q6&7 More Comprehensive Evaluation**.
>
> Thank you for your suggestions. We include two additional numerical experiments here.
>
> **Compare our VPT prior with learnable histogram (LH) prior**
>
> We set the number of bins of a LH to $K=2^L$ to match the number of leaves in a $L$ level VPT. For each dimension $d$, we parametrize a strictly increasing sequence $b_0^{(d)} < b_1^{(d)} < \cdots < b_K^{(d)}$, by setting the lower boundary $b_0^{(d)}$ and $b_k^{(d)} = b\_{k-1}^{(d)} + softplus(\delta^{d}\_{k-1})$. The parameters $\delta^{d}_{k-1}$ are learned jointly with the backbone network, and $\rm softplus(\cdot)$ guarantees positive bin widths.
>
> Given one dimension $x^{(d)}$, we can find the active bin index $k_*$, then compute the log-density as
> \begin{align*}
>     \log p(x^{(d)}) = \log p_{k_*}^{(d)} - \log (b_{k_*+1}^{(d)} - b_{k_*}^{(d)}).
> \end{align*}
> The overall log density of learnable histogram (LH) can be computed as $\log p_{\rm LH}(x)= \sum^D_{d=1}\log p(x^{(d)})$.
>
> The following table shows the results on UCI datasets with Block-NAF backbone (same as the our paper). The learning histogram prior is used as a plug-in here. All the hyper-parameters are kept the same as those in VPT.
>
> **Table— Comparison of learnable histogram (LH) prior and VPT prior**.  Results are averages over 5 runs (mean ± std).
> | Dataset | POWER | GAS | BSD300|
> |---------|-------|-----|--------|
> |LH|0.62_(0.01)|11.00_(0.20)| 156.98_(0.93)|
> |VPT| 0.67_(0.01) |11.92_(0.10)| 158.59_(1.02)|
>
> From the results we can observe a clear advantage of VPT over LH, although they share the same number of parameters. Besides the results on UCI data, we also compare LH with VPT on CIFAR-10 with NICE network (same in the paper). With $L=4$, our VPT obtains a result of $2.06$ bits per dim (BPD), and LH obtain $2.57$ BPD.
>
> VPT uses a Bayesian-nonparametric Pólya-tree prior, coupling every Beta node to its ancestors. Evidence at coarse levels shrinks finer splits, so sparsely populated bins “borrow” strength from their parents and avoid over- or under-fitting. Learnable histograms lack this hierarchy and often overfit low-density regions or let empty bins collapse.
>
> Unlike MAP-learned histograms, VPT’s variational inference keeps the joint posterior over all splits, yielding a coherent, uncertainty-aware density— even with a truncated tree. The hierarchy also provides multi-scale probability mass, offering intuitive coarse-to-fine insights alongside robust density estimates.
>
> **Compare the uncertainty calibration with other uncertainty estimation methods**
> The following is our experimental settings including the evaluation metrics and descriptions of comparable methods.
>
> **Metric**: Standardized squared error (SSE).
> For every test example and every latent  dimension $d=1,\ldots,D$ we compute
> $$
>  z_{d}^{(n)} =
> \frac{\bigl(x_{d}^{(n)}-\mu_{d}^{(n)}\bigr)^{2}}
>           {\sigma_{d}^{2\,(n)}},
>    \text{SSE}=\frac1{ND}\sum_{n,d} z_{d}^{(n)}.
> $$
> If the predictive variance is perfectly calibrated, $\mathbb E[z]=1\Rightarrow {SSE}=1$; SSE $>1$ indicates under‐estimated variance (model over-confidence); SSE $<1$ indicates over‐estimated variance (model under-confidence).
>
> **Baselines**: We currently benchmark against two standard uncertainty methods
> - MC-dropout with a  rate of $0.15$ \[3\].
>
> - Variational BNN with mean-field Gaussian weights \[4\].
>
> [3] Yarin Gal etc. Dropout as a Bayesian Approximation. ICML 2016
> [4] Radford M. Neal. Bayesian Learning for Neural Networks. Ph.D. Thesis, 1995.
>
> |Dataset|VPT|MC-Dropout|BNN|
> |---------|-----|-----------|-----|
> |POWER|0.92|1.30|1.17|
> |GAS|0.90|1.35|1.14|
> |MNIST|0.92|1.27|1.08|
>
> The table shows that VPT slightly over-estimates variance yet achieves the best calibration. MC-Dropout and BNN both under-estimate variance (BNN fares slightly better than MC-Dropout). We think VPT’s hierarchical shrinkage inflates variance in sparse regions, whereas MC-Dropout and mean-field BNNs fail to propagate uncertainty fully, leading to systematic under-confidence.
>
> We also emphasize that VPT’s variances are analytic, no Monte-Carlo sampling is required at test time—whereas both baselines need $30$ forward passes per input. Thus VPT offers better calibration at virtually zero extra compute.
>
> We will incorporate this calibration study—including additional datasets and seeds—into the revised paper to substantiate our uncertainty claims even more thoroughly.
>
> We appreciate the reviewer’s thoughtful feedback. Our new results show that VPT scales well to larger models and highlight the benefits of its closed-form VI over other approaches. We have also added two new numerical studies, which will appear in the revision. We hope these clarifications fully address your concerns and it is much appreciated if you could consider raising your score.

---

> > ### Comment · Reviewer_vPbf · 2025-08-03
> >
> > Thank you for your good reply! In Q3 I mean VPT is biased. Sorry for the typo. One of the biggest drawbacks for variational methods is biasedness, making it faster. VPT combines, however, seemingly, the drawbacks of the two. It is slow, but biased.

---

> > > ### Author Response · Authors · 2025-08-03
> > >
> > > We are glad that you find our response satisfying. And thank you for your clarification on Q3. We agree that variational methods can introduce bias compared to exact estimators, but because our VPT objective is based on the closed-form joint posterior likelihood (thanks to Polya tree conjugacy), it is less biased than typical mean-field VI approaches, which optimize only a loose bound. Moreover, unlike many modern VIs that rely on expensive Monte Carlo estimates of the ELBO, our closed-form likelihood means we do not need to sample during optimization—so we retain the speed and stability of traditional VI while achieving more accurate posterior approximations than mean-field baselines. Thank you again for your great efforts on our manuscript!

---

### Note · Authors · 2025-08-12

We sincerely thank all reviewers for their thoughtful and constructive feedback. We are encouraged by the consensus that our work, bridging Bayesian nonparametrics and modern deep learning, is both solid and promising. We greatly appreciate the recognition that our variational Pólya tree (VPT) provides a principled and efficient approach to modeling uncertainty of the parameter estimates. By leveraging the conjugacy and hierarchical structure of the Pólya tree, VPT avoids mean-field independence assumptions while remaining practical for deep learning applications. We are also pleased that the reviewers found our experiments, which spans multiple domains, comprehensive and impactful. Additionally, the tree structure offers clear interpretability while introducing only negligible parameter and memory overhead.


To address the main concerns raised, we have added a learnable-histogram ablation study that isolates the effect of VPT’s structural regularization, as well as a quantitative uncertainty-calibration study showing that our method delivers well-calibrated variances without extra computational cost. These additions provide robust, quantitative evidence of VPT’s advantages. We are confident that planned refinements to the writing and presentation will further highlight these contributions.

Once again, we thank the reviewers for their time, effort, and constructive engagement, and we promise to incorporate these improvements into the final version.

---

### Decision · Program_Chairs · 2025-09-17

**Decision:**

Accept (poster)

**Comment:**

The reviewers agree this paper has many interesting ideas: Polya trees are not as popular as other BNP methods in the NeurIPS community, the idea of fusing ideas between Bayesian nonparametrics and deep learning is interesting. However, there were also a number of concerns raised, listed below. Overall though, the reviewers believe that these can be addressed while preparing the camera ready version. Please look at these and well as you discussion with the reviewers when preparing the camera ready version:

1) Clarity: The presentation of Polya trees is very brief and does not clearly articulate their benefits (e.g. obscuring the usefulness of the proposed variational approximation). Additionally, it seems to have a number typos/unclear notation. E.g. what is the difference between subscripts and superscripts for the betas?  I also think the paper oversells Polya trees a little. It mentions that Dirichlet processes "induce discreteness, making them suboptimal for directly modeling continuous distributions". Of course DP mixture models fix this. By contrast, the Polya tree approach here seems truncated at level L, which introduces its own issues. I think this warrants more discussion.
3) Claims vs results about interpretability and uncertainty quantification: The first half of the paper talks about the interpretability of the proposed method about 5 times, but the presented results are underwhelming. Reviewers were also not very satisfied with claims about uncertainty quantification. With their rebuttal, the authors left the reviewers feeling much better about these points, so please incorporate the latter into the final version
3) Scalability and empirical evaluation: this should be discussed more thoroughly, especially since this paper focuses (without being clear about this) on univariate Polya trees